# Asset exposure data for global physical risk assessment

Samuel Eberenz[1,2], Dario Stocker[1,2], Thomas Röösli[1,2], David N. Bresch[1,2]

[1] Institute for Environmental Decisions, ETH Zurich, Zurich, 8092, Switzerland
[2] Federal Office of Meteorology and Climatology MeteoSwiss, Zurich-Airport, 8058, Switzerland

*Correspondence to*: Samuel Eberenz (eberenz@posteo.eu)

**Abstract.** One of the challenges in globally consistent assessments of physical climate risks is the fact that asset exposure data are either unavailable or restricted to single countries or regions. We introduce a global high-resolution asset exposure dataset responding to this challenge. The data are produced using "lit population" (LitPop), a globally consistent methodology to 10 disaggregate asset value data proportional to a combination of nightlight intensity and geographical population data. By combining nightlight and population data, unwanted artefacts such as blooming, saturation, and lack of detail are mitigated. Thus, the combination of both data types improves the spatial distribution of macroeconomic indicators. Due to the lack of reported subnational asset data, the disaggregation methodology cannot be validated for asset values. Therefore, we compare disaggregated GDP per subnational administrative region to reported gross regional product (GRP) values for evaluation. The 15 comparison for 14 industrialized and newly-industrialized countries shows that the disaggregation skill for GDP using nightlights or population data alone is not as high as using a combination of both data types. The advantages of LitPop are: global consistency, scalability, openness, replicability, and low entry threshold. The open-source LitPop methodology and the publicly available asset exposure data offer value for manifold use cases, including globally consistent economic disaster risk assessments and climate change adaptation studies, especially for larger regions yet at considerably high resolution. Code is 20 published on GitHub as part of the open-source software CLIMADA (CLIMate ADAptation) and archived in the ETH Data Archive with link: http://doi.org/10.5905/ethz-1007-226 (Bresch et al., 2019b). The resulting asset exposure dataset for 224 countries is archived in the ETH Research Repository with link: https://doi.org/10.3929/ethz-b-000331316 (Eberenz et al., 2019).

## 1 Introduction

The modelling of climate risks on a global scale requires globally consistent data representing hazard, vulnerability, and exposure, as defined by the Intergovernmental Panel on Climate Change (IPCC, 2012, 2014) among others. While natural hazard data can be derived from general circulation models, there is a lack of consistent exposure data on a global scale. Exposure data is frequently defined as an inventory of elements at risk from natural hazards (Cardona et al., 2012; UNISDR, 2009). For the modelling of physical risk as the direct economic impacts of disasters, exposure should specifically represent 30 the spatial distribution of physical asset stock, i.e. buildings and machinery. While aggregate estimates of asset values are available at country level, open data on the spatial distribution of asset values are scarce. Proprietary asset exposure data (e.g. owned by insurance companies) are usually not publicly available.

Due to the lack of comprehensive asset stock inventories, large scale asset exposure maps are often estimated top-down, using downscaling techniques (De Bono and Mora, 2014; Gunasekera et al., 2015; Murakami and Yamagata, 2019). On a country 35 aggregate level, estimates of total asset values can be derived from socioeconomic flow measures, such as gross domestic product (GDP), since the two indicators exhibit strong correlations (Kuhn and Ríos-Rull, 2016). Annual values of socioeconomic flow variables, particularly GDP, are often more readily available than asset values. Assuming that human presence and activity are proxies of economic output, downscaling of GDP has been based on geographical population data (Kummu et al., 2018) and on population combined with land-use, road networks, and locations of airports (Murakami and

Yamagata, 2019). High resolution yearly GDP maps based on these approaches are publicly available (Geiger et al., 2017; Kummu et al., 2018). Global asset exposure data were produced for the Global Assessment Report 2013 of the United Nations Office for Disaster Risk Reduction (UNISDR), following a downscaling approach (De Bono and Mora, 2014). However, the data's use beyond the scope of the Global Assessment Report is limited, because the data represents urban areas only and the methodology is not easily reproducible and thus not adaptable. For future quantitative risk assessments, more recent exposure

data would be desirable. An alternative methodology to model global asset exposure based on the combination of diverse datasets was presented by Gunasekera et al. (2015). The authors combined data on built-up area, building typologies, and construction cost with sector specific asset data and GDP disaggregated proportional to population density. Unfortunately, the source code and resulting exposure data have not been made publicly available. Reproducing these previously mentioned exposure modelling efforts is beyond the scope of most economic disaster risk assessments and climate change adaptation

studies.

In recent years, the use of nightlight intensity from satellite imagery has seen a marked increase in usage in science in general and especially for the disaggregation of socioeconomic indicators (Elvidge et al., 2012; Gettelman et al., 2017; Ghosh et al., 2013; Mellander et al., 2015; Pinkovskiy, 2014; Sutton et al., 2007; Sutton and Costanza, 2002). Being publicly available and updated regularly, global nightlight images have been proven to be a useful source of information and is commonly used in

scientific contexts for the estimation of unavailable GDP or growth data (Henderson et al., 2012). However, there are some technical limits to the usage of nightlight satellite imagery (Han et al., 2018), especially saturation and blooming. As luminosity can only be distinguished up to a certain brightness, saturation may lead to very bright spots being underrepresented. In state-of-art nightlight products from the Suomi National Polar-orbiting Partnership's Visible Infrared Imaging Radiometer Suite (VIIRS), there are 256 shades of brightness, from the minimum zero (no light emission) to the maximum 255 (NASA Earth

Observatory, 2017; Román et al., 2018). Any pixel brighter than what would entail a value of 255 will also appear at this value (Elvidge et al., 2007). Brightness can exude from bright pixels to neighboring pixels, causing the brightness in the latter to be overestimated, leading to blooming. This issue occurs especially in large urban areas and on specific surfaces, such as sand and water (Elvidge et al., 2004; Small et al., 2005). As a consequence of saturation, socioeconomic indicators scale rather exponentially than linearly with nightlight intensity (Sutton and Costanza, 2002; Zhao et al., 2015, 2017). To counteract the

saturation effect, Gettelman (2017) and Aznar-Siguan and Bresch (2019) used exponentially scaled nightlight intensity as a basis for GDP disaggregation for tropical cyclone risk assessments. Saturation and blooming can also be mitigated by combining nightlights with other data types: Sutton et al. (2007) combined the areal extend of lit area with population data to estimate GDP at a subnational level. Zhao et al. (2017) enhanced nightlight intensity values with population data to get a more accurate estimation of spatial economic activity in China. This is based on the observation that there is also an exponential

relationship between nightlight intensity and population density. The authors showed that the product of nightlight intensity and gridded population count (called "lit population" by the authors), is a better proxy for economic activity in China than nightlight intensity alone.

Here, we are using and expanding the "lit population" approach presented by Zhao et al. (2017) to define and implement a globally consistent methodology for asset exposure disaggregation, named LitPop hereafter. This paper presents global gridded

asset exposure data, and documents and evaluates the underlying LitPop methodology. The resulting asset exposure dataset for 224 countries is made available online at the ETH Research Repository (Eberenz et al., 2019). It is suitable to provide the globally consistent asset exposure base for modelling physical risks. The methodology is published on GitHub as part of the open-source event-based probabilistic impact model CLIMADA (CLIMate ADAptation) (Aznar-Siguan and Bresch, 2019; Bresch et al., 2019a) and archived in the ETH Data Archive (Bresch et al., 2019b).

Information on input data, methodology, and the evaluation approach are provided in Section 2. Subsequently, the resulting global asset exposure data are presented and evaluation results shown in Section 3. The advantages and limitations of the methodology are discussed in Sections 4. Please refer to section 5 for data and code availability.

## 2 Data & Methods

### 2.1 Overview

The core functionality of the LitPop methodology is the spatial disaggregation of national total asset values to obtain a gridded asset exposure product. Gridded nightlight intensity (Section 2.2) and gridded population data (Section 2.3) are combined to compute a digital number at grid cell level. Physical asset stock values (i.e. produced capital, Section 2.4.1) are then disaggregated proportional to the digital number per grid cell (Section 2.5). This results in the gridded asset exposure dataset presented here. Instead of the physical asset stock, GDP (Section 2.4.2) or gross regional product (GRP, Section 2.4.3) can be

distributed to obtain GDP per grid cell. Because of a lack of subnational produced capital data, GDP and GRP are used to evaluate the methodology by assessing the subnational disaggregation skill for varied combinations of the input data, as described in Section 2.6. A detailed overview over the input data is provided in Table 1 the disaggregation approach is illustrated in Figure 1.

| Input data | Usage | Spatial resolution | Reference year | Data Source | Description |
|---|---|---|---|---|---|
| Gridded nightlights (Lit) | Disaggregation | 15 arcsec | 2016 | NASA's Black Marble nighttime lights (NASA Earth Observatory, 2017; Román et al., 2018) | Section 2.2 |
| Gridded population (Pop) | Disaggregation | 30 arcsec (224 countries) | 2015 | Gridded Population of the World (GPW) (Center for International Earth Science Information Network (CIESIN), 2017) | Section 2.3 and Table S1 |
| Produced capital | Estimation of total asset value | 140 countries | 2014 | World Bank Wealth Accounting (World Bank, 2019a) | Section 2.4.1 and Table S1 |
| GDP-to-wealth ratio | Estimation of total asset value | 84 countries | 2017 | Global Wealth Report (Credit Suisse Research Institute, 2017) | Section 2.4.1 and Table S1 |
| GDP | Estimation of total asset value and evaluation | 224 countries | 2014* | World Bank Open Data portal (World Bank, 2019b) | Section 2.4.2 and Table S1 |
| GRP | Evaluation | 507 regions in 14 countries | 2012-2017 | Various sources, c.f. Table A1 | Section 2.4.3 and Table A1 |

**Table 1: Overview of input dataset, including information on usage, resolution, reference year, data source, and references. The**

**reference year indicates the year for which the data used was provided. *) For GDP, the value of 2014 in current USD was used for 203 countries. For 24 countries without GDP data available for 2014, closest available data points from the years 2000 to 2017 were used instead.**

## 2.2 Satellite nightlight data

The nightlight intensity product used here are nighttime lights of the Black Marble 2016 annual composite of the VIIRS day-night band (DNB) at 15 arcsec resolution (Román et al., 2018), downloaded from the NASA Earth Observatory (2017). The processed datasets of luminosity by human activity based on VIIRS mark an distinct improvement over previous technologies, allowing for a greater range of light to be recorded (Carlowicz, 2012). The sun-synchronous satellite passes each place on Earth twice a day, at approximately 01:30 and 13:30 local time. Nightlight intensity on a scale from 0 to 255 is a variable
derived from raw measurements. To isolate luminosity from sustained human activity, the Black Marble nightlight product includes corrections for Lunar artefacts, cloud, terrain, atmospheric, snow, airglow, stray light, and seasonal effects (Carlowicz, 2017; Lee et al., 2014; Román et al., 2018). The data is provided for 2012 and 2016 at a resolution of 15 arcsec, which corresponds to around 500 m at the equator. The open-source code developed here can be adapted easily to use other versions and sources of nightlight data. This could be of interest for near-time applications in the future, as daily nightlight images
could be available in the future (Carlowicz, 2017).

## 2.3 Gridded population data

The Gridded Population of the World (GPW) dataset is a spatially explicit representation of the world's population. It is based on two sets of inputs: non-spatial population data and cartography data. Using census data or population figures by the official national statistics offices, it uniformly distributes the numbers at the smallest available administrative unit to the corresponding
cartographic shape, without taking into account any ancillary sources (Doxsey-Whitfield et al., 2015). The data quality for each country strongly depends on the underlying level of availability of population data. For example, for Canada, population data is available down to the fifth subnational administrative unit, of which 493'185 exist. The information for Canada is hence a lot more fine-grained than for instance for Jamaica or Uzbekistan, where population numbers are only recorded at the first subnational administrative unit (Socioeconomic Data and Applications Center (SEDAC), 2017). The level of detail and number
of subnational administrative units resolved per country are listed in Table S1. While modelling is kept at a minimum in the GPW dataset, values are inflated or deflated from the latest year with data available to 2000, 2005, 2010, 2015, and 2020 (Center for International Earth Science Information Network (CIESIN), 2017).

GPW was selected for the LitPop methodology because, unlike other spatial population datasets, it does not incorporate nightlight satellite data or other auxiliary data sources (Leyk et al., 2019). This allows us to enhance nightlight data with a
completely independent dataset. Moreover, it is released under the creative commons license. From GPW, the Population Count v4.10 data at the highest available resolution, 30 arcsec, is used, because it is the closest to NASA's nightlight dataset, both in terms of spatial resolution and available time steps.

## 2.4 Socioeconomic indicators

### 2.4.1 Total asset value per country

The World Bank's produced capital stock (World Bank, 2018) is one of the most comprehensive global estimates of the value of manufactured or built assets per country. It has been used as an indicator of exposure to natural disaster in the UNISDR's Global Assessment Report 2013 (De Bono and Mora, 2014). Produced capital accounts for machinery, equipment, and physical structures (World Bank, 2018). It also includes a fixed scale-up of 24% to account for the value of built-up land.

Produced capital values are currently available for 140 countries and 5 time steps: 1995, 2000, 2005, 2010, and 2014 from the
World Bank Wealth Accounting (World Bank, 2019a). Per default, the scale-up for built-up land is subtracted, assuming that there is no direct damage to the value of the land itself in the case of disaster. While not universally true, this assumption is based on the focus of the asset exposure data for the purpose of assessing direct impact to tangible structures. For applications considering the impact on the value of land, the linear scale-up can be reapplied before utilization of the asset exposure data.

Out of a total of 250 countries we considered for the production of this dataset, produced capital numbers for 2014 are available
for 140 countries. For these 140 countries, produced capital for 2014 was used here as total asset value for disaggregation. For additional 87 countries, total asset values were set to non-financial wealth. Non-financial wealth was computed from the country's GDP and the GDP-to-wealth ratio estimates derived from the Credit Suisse Research Institute's Global Wealth Report (Credit Suisse Research Institute, 2017). This approach has previously been followed by Geiger et al. (2018). We compared produced capital and non-financial wealth for 140 countries (Table S1) and found that non-financial wealth can be used as a
conservative approximation of produced capital. For 59 of the 87 countries without produced capital data, an average GDP-to-wealth ratio of 1.247 was applied. In summary, the whole dataset contains gridded asset exposure data for a total of 224 countries, ignoring 26 countries and areas due to lack of data. Missing countries and areas (with currently assigned ISO 3166-1 alpha-3 codes) are Aland Islands, Antarctica, Bonaire, British Indian Ocean Territory, Sint Eustatius and Saba, Bouvet Island, Cocos (Keeling) Islands, Christmas Island, Guadeloupe, French Guiana, French Southern Territories, Heard Island and
McDonald Islands, Holy See, Kosovo, Libya, Martinique, Mayotte, Pitcairn, Palestine, Reunion, South Georgia and the South Sandwich Islands, South Sudan, Svalbard and Jan Mayen, Syrian Arab Republic, Tokelau, United States Minor Outlying Islands, and Western Sahara. An overview over the utilized data per country, including, produced capital (were available), GDP-to-wealth ratios, and GDP for 2014 is provided in Table S1.

## 2.4.2 GDP

GDP is a well-established indicator of macroeconomic output. For most countries in the world, annual values are available dating back several decades. National GDP data in current USD of 2014 or the nearest available year are retrieved from the World Bank Open Data portal (World Bank, 2019b).

While GDP is not a direct measure of physical asset values, it is used here both for scaling asset values in time to fill data gaps and for the evaluation of the LitPop methodology. The underlying assumption is that within a country, GDP and wealth are
correlated, i.e. a higher GDP value is equivalent to higher asset values. This correlation has been established in empirical studies (Kuhn and Ríos-Rull, 2016).

## 2.4.3 GRP

The subnational equivalent to GDP is often referred to as GRP. GRP can be used to improve the downscaling of GDP, especially for countries with considerable regional differences. As described in Section 2.6 below, we use GRP data from 14
countries to evaluate the LitPop methodology by assessing its skill to disaggregate national GDP to a subnational level. As there is no unified data source for GRP, it was gathered manually from government sources and OECD.Stat (Organisation for Economic Co-operation and Development, 2019). The countries used for validation are Australia, Brazil, Canada, Switzerland, China, Germany, France, Indonesia, India, Japan, Mexico, Turkey, USA, and South Africa. The aim of the selection was to include countries from an as wide as possible range of income groups and world regions. Since the selection of countries was
limited by the availability of GRP data, the selection has a bias towards industrialized and newly industrialized OECD member states. According to World Bank income groups, these countries include eight countries from the high-income group (World Bank income group 4), four countries from the upper-middle-income group (3), two countries from the lower-middle-income (2), and no countries from the low-income group (1). Income groups and data sources per country are listed in Table A1 in the Appendix.

**2.5 Disaggregation of asset exposure**

To produce a high-resolution asset exposure map, the total asset value per country is disaggregated proportional to a function of nightlight luminosity and population count. This approach is closely adapted from the work of Zhao et al. (2017). In their paper, historic GDP is disaggregated proportionally to a digital number computed from a multiplicative function of nightlights and population with the aim to make spatial GDP predictions for China. The underlying idea is to enhance brightness values

with spatial population data to get a more accurate estimation of spatial economic activity. The work flow of the asset exposure disaggregation is described here in detail and illustrated in Figure 1.

In a first step, the two gridded input datasets are interpolated linearly to the same resolution of 30 arcsec. Then, the combination of the two aforementioned datasets is conducted for each grid cell:

$$Lit^n Pop^m{}_{pix} = \left(NL_{pix} + \delta\right)^n \cdot Pop_{pix}{}^m \tag{1}$$

Where the digital number value $Lit^m Pop^n{}_{pix}$ per grid cell ($pix$) is computed from the grid cell's nightlight intensity $NL_{pix} \in [0, 255]$, population count $Pop_{pix} \in \mathbb{R}^+$, as well as the exponents $n, m \in \mathbb{N}$. For all $m > 0$, the added $\delta$ is equal to 1 to ensure that non-illuminated but populated grid cells do not get assigned zero value. In the case that nightlight data is used on its own without population data ($m = 0$), $\delta$ is set to 0.

In a second step, gridded $Lit^m Pop^n$ is taken as a relative representation of economic stocks at each grid cell. It is used to linearly
disaggregate a total asset values of a country to a geographical grid. More precisely, the value of $Lit^m Pop^n{}_{pix}$ relative to the sum of $Lit^m Pop^n$ over all pixels within the boundaries of the country determines how much of a total value is assigned to each grid cell:

$$I_{pix} = I_{tot} \cdot \frac{Lit^n Pop^m{}_{pix}}{\sum_{pix\_i}^{N}\left(Lit^n Pop^m{}_{pix\_i}\right)} \tag{2}$$

Where $I_{pix}$ denotes the asset value per grid cell. The given value of a country's total asset value $I_{tot}$ is distributed to each grid
cell $pix$ proportionally to the $Lit^m Pop^n$-share of the grid cell. $N$ denotes the total number of grid cell (iterator $pix\_i$) inside the boundaries of the country.

Changing the exponents $m$ and $n$ determines with which power the two input variables contribute to the disaggregation function. The exponents $m$ and $n$ do not only weight relatively between $Lit$ and $Pop$ but they also determine the contrast in the distribution between all grid cells within a country. The larger the exponent, the more value is concentrated on grid cells with
large values of $Lit$ or $Pop$ respectively. The aim of the evaluation described in Section 2.6 is to compare disaggregation skill of varied combinations of $m$ and $n$ and select the most adequate combinations for subnational disaggregation.

For the creation of the asset exposure data presented here, $I$ represents asset value, i.e. produced capital or non-financial wealth disaggregated per grid cell and $m$ and $n$ set to 1. For the evaluation of the disaggregation skill of the approach presented in the following section, $I$ represents the flow variable GDP instead, as in the study of Zhao et al. (2017).

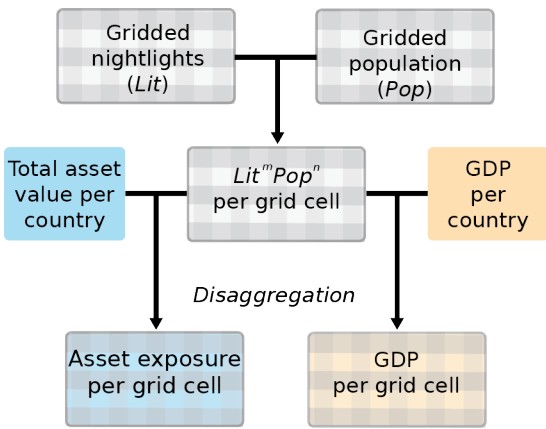


**Figure 1: Work flow of the LitPop downscaling: Gridded nightlights (*Lit*) and population (*Pop*) data are combined to compute gridded digital number *Lit^m Pop^n* (Eq. 1). Then, total asset value per country (i.e. produced capital or non-financial wealth) is disaggregated proportional to *Lit^m Pop^n* to obtain gridded asset exposure data (Eq. 2). GDP is disaggregated in the same way and compared against reported GRP for the evaluation of the downscaling approach.**

**2.6 Evaluation**

Gridded population and nightlight intensity can both be used as proxies for the spatial distribution of asset exposure. Both proxies have limitations: an asset-distribution proportional to population density assumes that physical wealth is distributed equally among the population and that assets are located exactly where people live. As already mentioned in Section 2.3, for many developing countries, gridded population data has a coarse resolution. Nightlight-based models, on the other hand, are
mainly limited by saturation and blooming as described in the Introduction. By combining nightlight intensity and population count, we expect to combine their skills while reducing the limitations mentioned above.

The LitPop approach's skill in disaggregating asset exposure cannot be assessed directly due to the lack of reference asset value data on a subnational level. Therefore, GDP and GRP are used instead for an indirect evaluation of the methodology. GDP and GRP are used to assess the subnational disaggregation skill, comparing varying combinations of the exponents $m$
and $n$ in $Lit^m Pop^n$.

The disaggregation skill is assessed as follows: (i) National GDP is disaggregated to grid level. (ii) The resulting gridded GDP is then re-aggregated for each subnational region (i.e. district, state, or canton) to obtain modelled GRP. (iii) Based on the comparison of normalized modelled and reported reference values of GRP, skill metrics are computed per country. In total, we use reported GRP data for 507 regions in 14 countries to evaluate the model's ability to distribute national GDP to
subnational regions.

To ensure comparability of skill metrics between different countries, GRP is normalized:

$$nGRP_i = \frac{GRP_i}{GDP} \tag{3}$$

Where $nGRP_i$ denotes the normalized GRP of subnational region $i$. Given that $GDP = \sum_i^N (GRP_i)$, it follows from Equation 3 that $\sum_i^N (nGRP_i) = 1$. Here, $N$ is the set of all subnational units in the country.

To assess the disaggregation skill per country, three skill metrics are computed from *nGRP*:

The Pearson correlation coefficient ρ (Equation 4) is computed to measure the linear correlation between the modelled *nGRP*$_{mod}$ and the reference value *nGRP*$_{ref}$. ρ is computed from the covariance (*cov*) and the standard deviations $\sigma_{mod} = \sigma(nGRP_{mod})$ and $\sigma_{ref} = \sigma(nGRP_{ref})$:

$$\rho = cov(nGRP_{i,mod}, nGRP_{i,ref})/(\sigma_{mod} \cdot \sigma_{ref}). \tag{4}$$

The correlation coefficient ρ is a widely used metric and straight forward to interpret and communicate: A value of 1 indicates a perfect positive linear correlation between the two variables while a value of 0 indicates that there is no linear correlation. However, ρ is no direct measure of the deviations of *nGRP*$_{mod}$ from *nGRP*$_{ref}$ and yields no information regarding the slope of the linear relationship. Therefore, it only represents a potential skill and needs to be evaluated in combination with a measure of the slope. The slope of the linear regression conveys the information, whether there is a systematic over- or underestimation
of economically large regions in the disaggregated data. $\beta = \rho \cdot \sigma_{mod}/\sigma_{ref}$ is calculated to complement the analysis: β larger (lower) than 1 implies an overestimation (underestimation) of the GRP of regions with relatively large GRP and an underestimation (overestimation) of regions with relatively low GRP by the downscaling within one country. Together, ρ and β allow for an evaluation of the linear fit between modelled and reference data.

Complementarily, the root-mean-squared fraction (RMSF) is a relative error metric, weighting the relative deviation for each
region equally, independently of the absolute values. Therefore, RMSF (Equation 5) puts equal weight to all subnational administrative units in a country, even if their GRP and thus their absolute difference between modelled and reference values are small. A RMSF of 1 indicates perfect fit. A RMSF-value of 2 means that on average, the modelled GRP deviates by a multiplicative factor of 2 from the reference value.

$$RMSF = exp\left(\sqrt{\frac{1}{N}\sum_i^N \left[\log\left(\frac{nGRP_{i,mod}}{nGRP_{i,ref}}\right)\right]^2}\right) \tag{5}$$

This analysis is applied using varying combinations of nightlight and population data for the disaggregation of GDP. The resulting skill metrics are compared for each combination and country.

## 3 Results

### 3.1 Global gridded asset exposure

We applied the LitPop methodology with the exponents *m = n = 1* to compute gridded asset exposure data for 224 countries
and areas worldwide (Fig. 2). Total physical asset values of 2014 were disaggregated proportionally to *Lit*$^1$*Pop*$^1$ to a grid with the spatial resolution of 30 arcsec (approximately 1 km). Total asset values in the dataset sum up to 2.51*10$^{14}$ (251 trillion) current USD of 2014. The 140 countries with produced capital data used as total asset value (c.f. Section 2.4.1), contribute USD 245 trillion (97.6 %) to the total asset exposure. The remaining 84 countries where asset values were estimated from GDP and a GDP-to-wealth ratio instead, contribute the remaining USD 6 trillion. In total, the 224 countries contribute around
99.9% to recorded global GDP. All numbers are based on the national values assembled in Table S1. Data sources are summarized in Table 1.

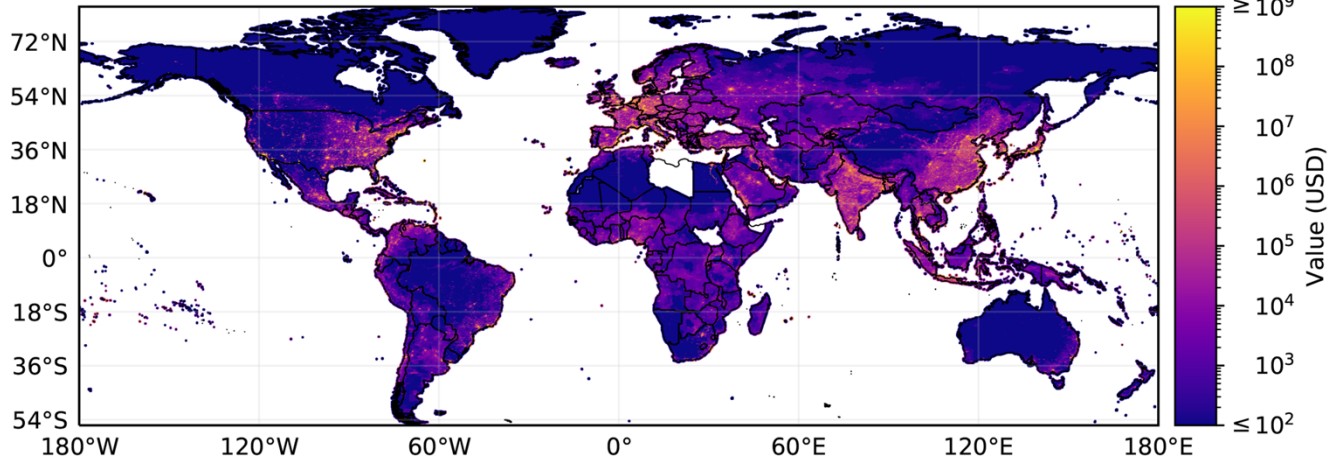

**Figure 2: World map showing gridded asset exposure values scaled to a resolution of 600 arcsec. The actual resolution of the underlying gridded data is 30 arcsec (~1 km). To obtain this dataset, national total asset values were disaggregated proportional to the distribution of $Lit^1Pop^1$ for 224 countries and areas. 26 countries and areas without data are left blank, including Libya, South Sudan, and Syria. The colormap is logarithmic and cropped at USD 100 (lower bound) and USD 1,000,000,000 (upper bound).**

In the following subsections, the LitPop methodology is evaluated both quantitatively and qualitatively: The results of the quantitative assessment of disaggregation skill introduced in Section 2.6 are presented in Section 3.2, providing justification for the selected combination of the exponents *m* and *n* for the global dataset. Differences between asset exposure distribution based on $Lit^1$, $Pop^1$, and $Lit^1Pop^1$ are shown by example of detail maps of two metropolitan areas (Section 3.3). Finally, limitations of the LitPop methodology are discussed by the example of GDP disaggregation in Mexico (Section 3.4).

### 3.2 Evaluation

To evaluate the performance of the LitPop methodology, we compute and compare the disaggregation skill in regards to GDP for varying exponents *m* and *n* in $Lit^mPop^n$ (Eq. 1 and 2). Here, we show the comparison based on 14 countries with a total of 507 regional GRP data points available. The 14 countries make up 67% (USD 168 trillion) of the total dataset's exposure and 64.5% (USD 52 trillion) of global GDP in 2014. Ten combinations of *m* and *n* are assessed: $Lit^1Pop^1$, $Lit^1$, $Lit^2$, $Lit^3$, $Lit^4$, $Lit^5$, $Pop^1$, $Pop^2$, $Lit^2Pop^1$, and $Lit^3Pop^1$. These exponent combinations were selected based on examples in the literature and then explored iteratively, stopping at combinations with decreased skill compared to lower order combinations. For each country and exponent combination, the median and the spread of three skill metrics are compared: ρ, β, and RMSF (Fig. 3 and Tables A2 and A3).

For ρ (Fig. 3a), $Lit^1Pop^1$ shows the best overall median of ρ (0.94) with the lowest interquartile range (IQR) of 0.09. The IQR is used here as a measure of variability of the skill metrics, as it signifies the difference between the 25th and the 75th percentile of the resulting skill metric. The same holds for β of $Lit^1Pop^1$ (median=1.03, IQR=0.12, Fig. 3b). In contrast, β is on average well below 1 for combinations exclusively based on *Lit* (i.e. $Lit^m$). A value of β below 1 indicates an underestimation of the GRP of economically larger regions compared and an overestimation of smaller regions. This can possibly be attributed to the saturation problem of nightlight intensity data, given that economically large regions usually accommodate more metropolitan areas where saturation occurs the most. This interpretation is supported by the relatively low values attributed to London and Mumbai metropolitan areas by $Lit^1$ shown in Section 3.3.

For purely population-based disaggregation, we found a median of β below 1 for *Pop¹* and well above 1 for *Pop²* (Fig. 3b). This suggests that disaggregation proportional to *Pop¹* underestimates the asset values in urban agglomerations, while it is overestimated by *Pop²*. For the metric RMSF, *Pop¹* (median=1.37, IQR=0.37) and *Lit⁴* (median=1.64, IQR=0.36) perform best, while *Lit¹Pop¹* has a median RMSF of 1.67 and an IQR of 1.29 (Fig. 3c).

Within the set of combinations exclusively based on *Lit* (n=0), the skill metrics β and RMSF perform best for *Lit⁴* (Fig. 3b,c), with median ρ improving for larger values of m, however changing little from *Lit⁴* to *Lit⁵* (Fig. 3a).

Based on the comparison of the disaggregation skill with varying exponent *m* and *n*, there are two candidates for the most adequate functionality: *Lit¹Pop¹* (best ρ and β) and *Lit⁴* (best RMSF and best performance for n=0). The skill metrics of linear regression, ρ and β, give a better representation of the disaggregation skill for the absolute values than RMSF which is based on the relative deviation per data point. Prioritizing a better distribution of total values over relative performance, we conclude that *Lit¹Pop¹* can be considered the most adequate combination of *Lit* and *Pop* for the subnational downscaling of GDP. For countries with a lack of highly resolved population data, alternative datasets could be produced based on *Lit⁴* alone.

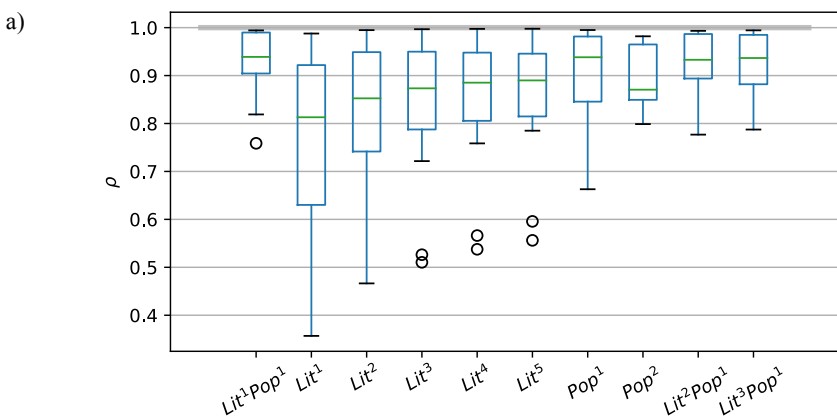

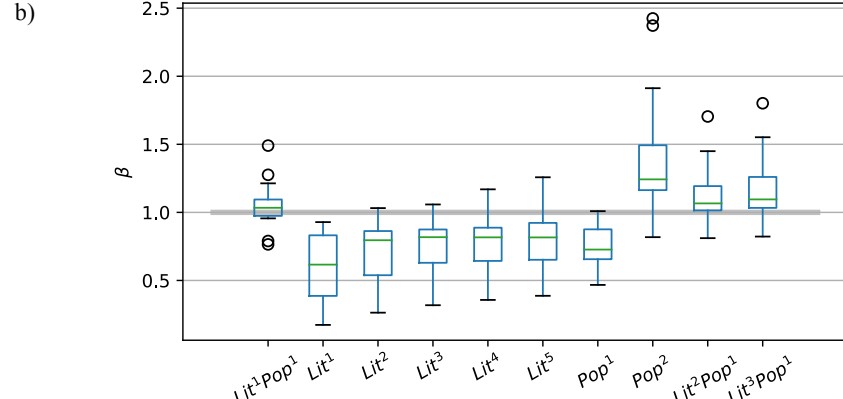

c)

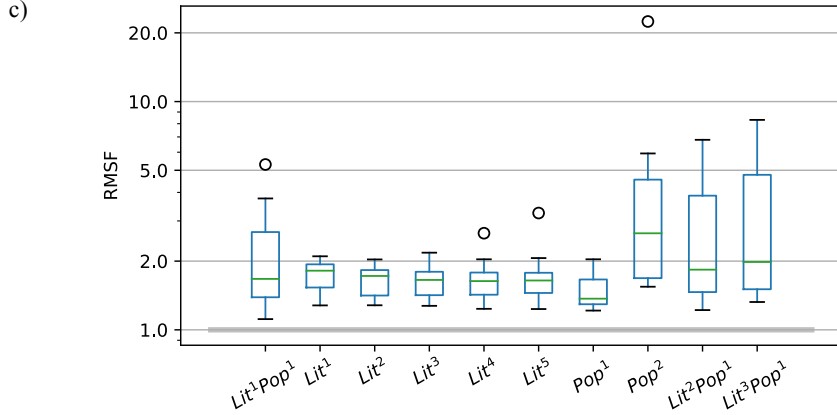

**Figure 3: Box plots showing the skill metrics ρ (a), β (b), and RMSF (c) for variations of *Lit^m Pop^n*. The metric value of 1, indicating perfect skill, is demarcated by the solid grey line. The plots are based on data from 14 countries and show the median (green), the 1st and 3rd quartile (IQR, blue box), data points outside the IQR but not more than 1.5\*IQR distance from the median (black whiskers), and outliers (black circles). RMSF is plotted on a logarithmic scale. Underlying metric values per country are listed in Table A2. Median and IQR per skill metric and combination of exponents are shown in Table A3.**

## 3.3 Detailed maps for metropolitan areas

Saturation and blooming in nightlight intensity data cause disaggregation based on nightlights alone to misrepresent actual value distribution, especially in urban areas. This can be seen in Figure 4, showing maps based on $Lit^1$ (a), $Pop^1$ (b) and $Lit^1 Pop^1$ (c). London (top row) and Mumbai (bottom), two large metropolitan areas, were chosen as examples. Comparable maps for Mexico City and New York are shown in Figure A1 in the Appendix.

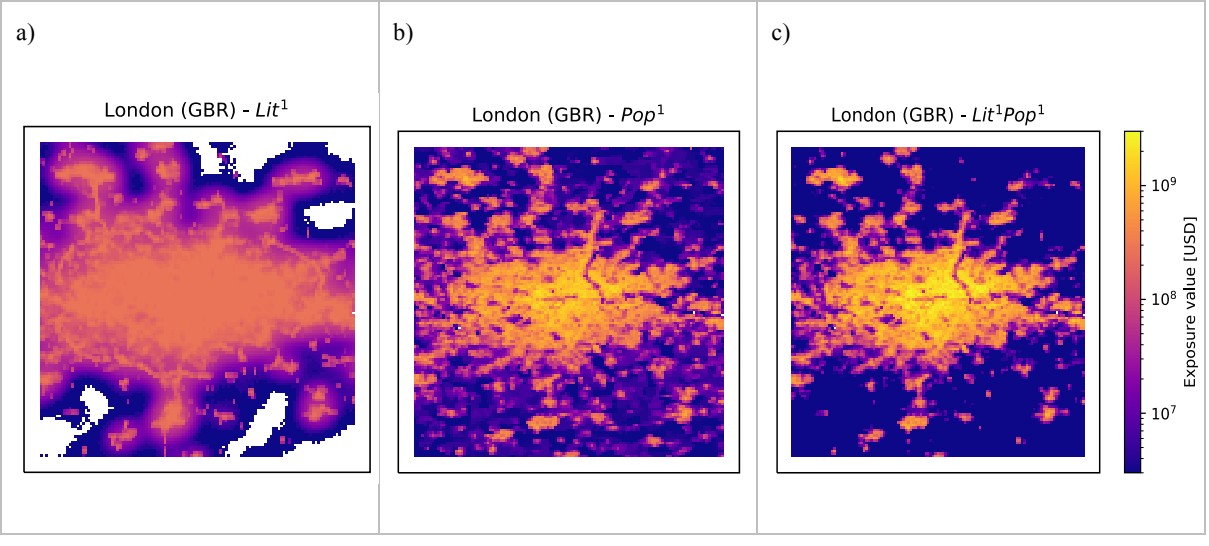

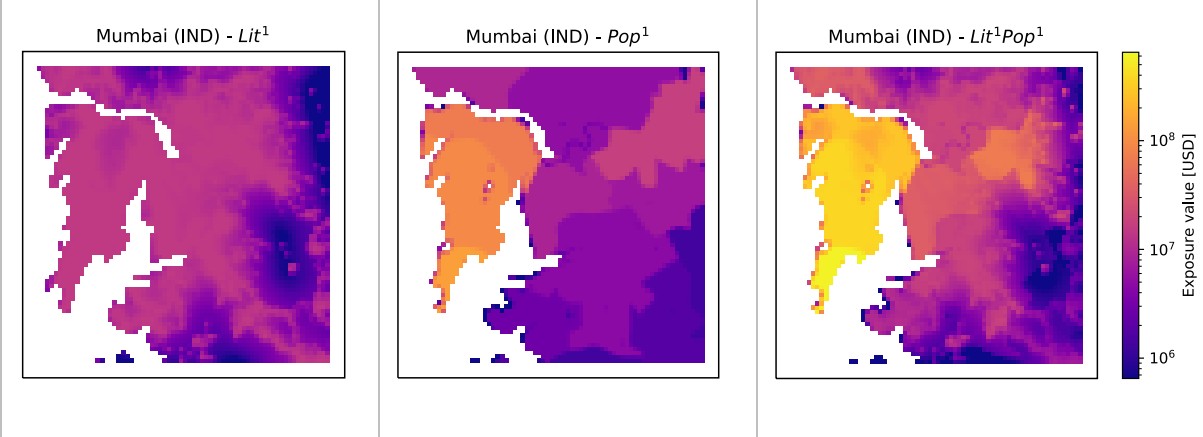

**Figure 4: Maps of disaggregated asset exposure value. Values are spatially distributed proportional to nightlight intensity of 2016 (*Lit¹*, a), population count as of 2015 (*Pop¹*, b), and the product of both (*Lit¹Pop¹*, c) for metropolitan areas in the United Kingdom (GBR) and India (IND). The maps are restricted to the wider metropolitan areas of London (0.6°W-0.4°E; 51-52°N) and Mumbai (72-73.35°E; 18.8-19.4°N) respectively. The colorbar shows asset exposure values in current USD of 2014 per pixel of approximately 1 km².**

The general exposure value level in the metropolitan areas shown in Fig. 4 are largest for $Lit^1Pop^1$ (Fig. 4c), highlighting a larger concentration of values in urban areas with this approach. The value distribution based on $Lit^1$ (Fig. 4a) does not show many details within the urban area. This effect is partially caused by saturation: the light radiation in the depicted areas is of such high intensity, that the nightlight data does not offer any way to distinguish different levels of human activity. We can also observe the blooming effect, with the luminosity of bright parts crowding out to neighboring pixels, causing them to appear brighter than their underlying light sources would warrant. This latter effect can be particularly illustrated over the Thames river and Bow Creek in the northeastern part of London: The unpopulated river area is resolved by $Pop^1$ (Fig. 4b top) but not by $Lit^1$ (Fig. 4a top). By taking population density into account, the $Lit^1Pop^1$ dataset enhances contrast and detail in urban areas (Fig. 4b, c). In addition, bright objects can be over-represented by $Lit^1$: In Figure 4a (top), the M25 London Orbital Motorway around London clearly stands out, with some pixels even at the same value as in central London.

As seen in the case of Mumbai, the $Lit^1Pop^1$ based asset exposure map of the metropolitan area in Figure 4c (bottom) shows much higher total values than those based on nightlights or population alone. This means that for $Lit^1Pop^1$, a larger proportion of the national produced capital of India is attributed to the metropolitan area of Mumbai as compared to $Lit^1$ and $Pop^1$ alone.

## 3.4 Example Mexico

The skill metrics for the subnational disaggregation of GDP in the country Mexico shows low values of ρ compared to most other countries for all tested values of *m* and *n* (ρ=0.76 for $Lit^1Pop^1$, c.f. Table A2a). The example of Mexico is presented here to illustrate limitations and uncertainties of the disaggregation approach. Figure 5 shows the data behind the evaluation for Mexico, i.e. modelled and reference *nGRP* for all 32 districts of Mexico. The corresponding plot data can be found in Table S2 as supplementary material. While the LitPop methodology performs well for most of the districts with relatively low GRP, it fails to reproduce reference *nGRP* for the main (capital) metropolitan region consisting of the districts México and Mexico City (Distrito Federal).

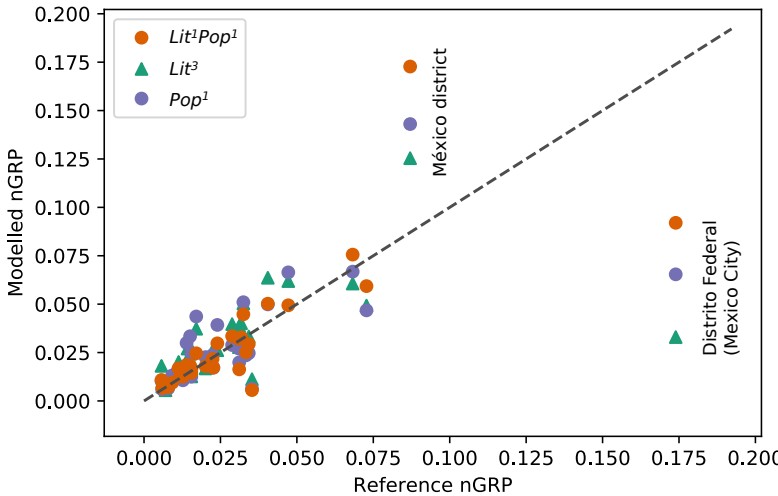

**Figure 5: Normalized gross regional product (*nGRP*) for the 32 districts of Mexico. Reference values are shown on the horizontal axis and modelled values on the vertical axis.**

The two districts with the largest GRP of the highly centralized country are Distrito Federal (Mexico City district) with a reference *nGRP* of 17.4% and México district (8.7%), surrounding the Distrito Federal. Asset exposure maps of the
330   metropolitan region are shown in Figure A1 in the Appendix. The disaggregation of GDP underestimates *nGRP* for Mexico City district while overestimating the value for México for all evaluated combinations of *m* and *n* (*nGRP* for *Lit$^l$Pop$^l$*, *Lit$^3$*, and *Pop$^l$* are shown in Figure 5). The overestimation of México district's *nGRP* indicates that the district has an over-proportional nightlight intensity and population count compared to a relatively low reference *nGRP*. Both districts combined sum up to modelled *nGRP* values of 11.2 to 17,6% for *Lit$^m$*, 20.8% for *Pop$^l$*, and 26.5% for *Lit$^l$Pop$^l$* (Table S2), the latter
335   agreeing well to a combined reference *nGRP* of 26.1%.

## 4 Discussion

The LitPop methodology allows for the creation of globally consistent and spatially highly resolved estimates of gridded asset exposure value. According to Pittore et al. (2017), efforts towards improving exposure data should aim at global consistency, continuous integration of new data and methods, and a careful validation of models and data. Here, we will discuss the
340   advantages and limitations of the LitPop methodology with regard to the following key criteria: Global consistency, disaggregation skill, scalability and flexibility, openness, replicability and reproducibility, and low entry threshold:

*Global consistency*. Based on globally available input data, the LitPop methodology was applied across countries from different continents and income groups. While the presented asset exposure dataset is not complete, it provides data for 224 countries contributing 99.9% of global GDP. Therefore, LitPop-based asset exposure data can be used as a basis for globally
345   comparable economic risk assessments. However, the evaluation of the of the methodology's disaggregation skill presented here is limited to an assessment of disaggregation skill for 14 OECD countries. It should be noted that due to lack of data we were not able to evaluate the method's performance for low income countries (World Bank income group 1). Therefore, the application of the asset exposure data for local assessments in countries within low income groups should be treated with

caution. Another caveat to global consistency is the fact that the quality and resolution of the underlying population dataset varies between countries, as discussed in greater detail in the next paragraph. As a consequence of these limitations, asset exposure data should be validated against local data before application for local risk assessments, especially in low income countries.

*Assessment of disaggregation skill.* For the gridded exposure dataset presented here, the LitPop methodology is used to disaggregate total asset values. Due to a lack of subnational reference asset values, the LitPop methodology's performance for the downscaling of asset stock values could not be evaluated directly. The assessment of disaggregation skill was instead based on the flow variables GDP and GRP. Given a correlation between stocks and flows within each country, this approach represents an indirect evaluation of the methodology for asset exposure downscaling. Evaluating 14 countries, we found that the LitPop methodology generally performs well in disaggregating GDP to subnational level. The skill metrics $\rho$ and $\beta$ showed that $Lit^1Pop^1$ distributes GDP better to the subnational level than the other combinations of nightlight and population data assessed. For RMSF, $Pop^1$ and $Lit^4$ perform best on average. We selected $Lit^1Pop^1$ as a basis for the disaggregated asset exposure dataset presented here. This decision is based on two considerations: (1) Giving $\rho$ and $\beta$ priority over RMSF because they are measures of absolute deviation between variables (as compared to RMSF that is a measure of relative deviation per data point); and (2) the fact that $Lit^1Pop^1$ combines the advantages of both input data types and mitigates their disadvantages, i.e. with regard to saturation, blooming, and detail. For countries without a high detail level in the population data available, asset exposure based on $Lit^mPop^n$ is more or less equivalent to one based on $Lit^m$ alone. For regional application in these countries, evaluation results suggest that disaggregation proportional to $Lit^4$ could distribute asset values best in the absence of detailed population data.

*Scalability and flexibility.* Subject to data availability, the LitPop methodology can be used to estimate the distribution of physical asset values for any target year at a wide range of resolutions. The data sources used here cater for resolutions up to 30 arcsec. While the GPW dataset provides population data for the previous two decades, the NASA nightlight images are currently only available for 2012 and 2016. The methodology includes a scaling of exposure data proportional to current GDP for years without any data available. The methodology can potentially be adapted to a variety of applications by an appropriate choice of the socioeconomic indicator that is disaggregated: The World Bank's produced capital data is used here as the default total asset value per country. Alternatively, GDP can be used as an estimator of economic output. GDP multiplied by a factor derived from the country specific income group can also be used to estimate asset values (Aznar-Siguan and Bresch, 2019; Geiger et al., 2017). This was done for countries without produced capital numbers available. Since the CLIMADA repository is open-source, the LitPop methodology can be amended to include alternative data sources and versions of both gridded nightlight, population, and total asset values, or other socioeconomic indicators to expand and update the asset exposure data. The LitPop methodology was developed to provide globally consistent asset exposure data for global-scale physical risk modelling. While it could be used for other applications as well, the limitations of its scope should be noted: The LitPop methodology does not account for differences in infrastructure types and vulnerability. In addition, gridded data may cause poor scoping of areas most vulnerable, or those with more exposed population. The example of Mexico (Section 3.4) illustrates the limitations of the LitPop methodology when it comes to the disaggregation of GDP within a metropolitan area: While the disaggregation of GDP proportional to $Lit^1Pop^1$ nicely reproduces the summed *nGRP* of the metropolitan area, the methodology fails to reproduce the distribution of *nGRP* between the two districts that make up the metropolitan area. Therefore, the use of the asset exposure data for local applications should be treated with care. The use for local or sector specific applications is limited without the addition of sector specific datasets. For risk assessments with a local focus as well as in countries of low income, we would advise to use more local approaches and bottom-up methods for identifying and analyzing the vulnerability component. Additionally, the asset exposure data could be further refined by including auxiliary data, such as road networks and land cover (Geiger et al., 2017; Murakami and Yamagata, 2019), or mobile phone cell antenna density (Brönnimann and Wintzer, 2018). In order to include sector specific assets not represented by the LitPop methodology, i.e. power plants or mines in unpopulated areas, additional sector specific asset inventories should be included (Gunasekera et al., 2015). For a globally consistent approach, sectoral data should however be included with caution, as such datasets are prone to regional or national biases.

*Openness, replicability, and low entry threshold.* The LitPop methodology was developed in the programming language Python 3 and published on the code hosting service GitHub as well as in a permanent repository (c.f. Section 5). The CLIMADA repository is developed open-source and makes use of open-access data to enable unrestricted use for applications also beyond academia. Next to the dataset provided, the LitPop-module can be used both to apply the computed asset exposure data for direct application in event-based risk assessments with CLIMADA or to export gridded asset exposure data to standard

formats for use in other applications. While $Lit^1Pop^1$ is the default, $Lit^mPop^n$ with custom exponents can be chosen as a basis for disaggregation. The documentation of CLIMADA is hosted on Read the Docs (https://climada-python.readthedocs.io/en/stable/). It includes an interactive tutorial of CLIMADA and the LitPop module (https://climada-python.readthedocs.io/en/stable/tutorial/climada_entity_LitPop.html), with guidance on how to compute and export LitPop based asset exposure data.

## 5 Data and code availability


Asset exposure data at a resolution of 30 arcsec for 224 countries, as well as normalized $Lit^1$ and $Pop^1$ for the 14 countries used for evaluation are archived in the ETH Research Repository with link: https://doi.org/10.3929/ethz-b-000331316 (Eberenz et al., 2019). The LitPop methodology is openly available as a module of CLIMADA (Bresch et al., 2019a) at GitHub under the GNU GPL license (GNU Operating System, 2007). CLIMADA v1.2.0 was used for this publication, which is

permanently available at the ETH Data Archive with link: http://doi.org/10.5905/ethz-1007-226 (Bresch et al., 2019b). The scripts reproducing the published dataset, as well as all figures in the present publication and the main results are published in the CLIMADA-papers repository on GitHub with link: https://github.com/CLIMADA-project (Aznar-Siguan et al., 2019).

## 6 Conclusion

The open-source LitPop methodology was developed to provide a geographical distribution of physical asset exposure values

that can be used to model first-order economic impacts of weather and climate events and other natural disasters. It integrates publicly available data sources to calculate gridded asset exposure estimates. The global consistency, flexibility and openness, and the integration in the CLIMADA repository offers value for manifold use cases for economic disaster risk modelling and climate change adaptation studies. However, the methodology could not be evaluated directly against subnational asset data and the evaluation based on GDP was limited to 14 OECD countries. Therefore, the asset exposure data is not suitable for

applications with a local or sector-specific focus without further validation. Future research and development could focus on the integration of higher resolved population data and other ancillary data sources as they become available globally. Validation against subnational asset value and empirical asset stock inventories yields the potential to evaluate and further improve the accuracy of asset exposure downscaling, both for global and regional applications. Regional validation could further inform the choice of the most appropriate downscaling functionality for different income groups and world regions.


## Appendix A

| Country | Regions | Income Group | Data Source | Reference year |
|---|---|---|---|---|
| Australia | 8 | 4 | Australian Bureau of Statistics, http://www.abs.gov.au/AUSSTATS/abs@.nsf/DetailsPage/5220.02016-17?OpenDocument | 2016 |
| Brazil | 27 | 3 | OECD.Stat, https://stats.oecd.org/ | 2015 |
| Canada | 14 | 4 | OECD.Stat, https://stats.oecd.org/ | 2016 |
| Switzerland | 26 | 4 | Swiss Federal Statistical Office, https://www.bfs.admin.ch/bfs/en/home/statistics/national-economy/national-accounts/gross-domestic-product-canton.assetdetail.6369918.html | 2014 |
| China | 31 | 3 | National Bureau of Statistics China, http://data.stats.gov.cn/english/easyquery.htm?cn=E0103 | 2015 |
| Germany | 16 | 4 | Statistische Ämter des Bundes und der Länder, https://web.archive.org/web/20110717065817/http://www.statistik-portal.de/Statistik-Portal/en/en_jb27_jahrtab65.asp | 2017 |
| France | 101 | 4 | Eurostat, http://ec.europa.eu/eurostat/web/regions/data/database | 2015 |
| Indonesia | 33 | 2 | OECD.Stat, https://stats.oecd.org/ | 2012 |
| India | 30 | 2 | Open Government Data Platform India, https://data.gov.in/catalog/capita-state-domestic-product-current-prices#web_catalog_tabs_block_10 | 2013/14 |
| Japan | 47 | 4 | Cabinet Office Government of Japan, http://www.esri.cao.go.jp/jp/sna/data/data_list/kenmin/files/contents/main_h26.html | 2014 |
| Mexico | 32 | 3 | National Institute of Statistics and Geography of Mexico, https://www.inegi.org.mx/sistemas/bie/?idserPadre=10200070#D10200070 | 2016 |
| Turkey | 81 | 3 | OECD.Stat, https://stats.oecd.org/ | 2014 |
| USA | 52 | 4 | US Bureau of Economic Analysis, https://www.bea.gov/data/gdp/gdp-state | 2016 |
| South Africa | 9 | 3 | OECD.Stat, https://stats.oecd.org/ | 2013 |

**Table A1: List of countries used for validation with the number of regions on the administrative level 1, the World Bank income group 2016, and GRP data source with URLs as accessed in January 2019. The income groups are: low income (1), lower middle income (2), upper middle income (3) and high income (4). In total, GRP data for 507 regions in 14 countries were used.**


| $\rho$ | AUS | BRA | CAN | CHE | CHN | DEU | FRA | IDN | IND | JPN | MEX | TUR | USA | ZAF |
|---|---|---|---|---|---|---|---|---|---|---|---|---|---|---|
| $Lit^1Pop^1$ | 0.99 | 0.98 | 0.99 | 0.94 | 0.93 | 0.90 | 0.92 | 0.90 | 0.82 | 0.93 | 0.76 | 0.99 | 0.98 | 0.99 |
| $Lit^1$ | 0.92 | 0.92 | 0.99 | 0.81 | 0.95 | 0.96 | 0.37 | 0.75 | 0.81 | 0.59 | 0.36 | 0.53 | 0.76 | 0.85 |
| $Lit^2$ | 0.93 | 0.96 | 0.99 | 0.89 | 0.96 | 0.94 | 0.47 | 0.79 | 0.82 | 0.73 | 0.47 | 0.66 | 0.78 | 0.95 |
| $Lit^3$ | 0.94 | 0.96 | 1.00 | 0.91 | 0.95 | 0.93 | 0.51 | 0.83 | 0.83 | 0.79 | 0.53 | 0.72 | 0.79 | 0.97 |
| $Lit^4$ | 0.94 | 0.97 | 1.00 | 0.93 | 0.95 | 0.92 | 0.54 | 0.85 | 0.84 | 0.82 | 0.57 | 0.76 | 0.80 | 0.97 |
| $Lit^5$ | 0.94 | 0.97 | 1.00 | 0.93 | 0.95 | 0.91 | 0.56 | 0.87 | 0.84 | 0.84 | 0.60 | 0.79 | 0.81 | 0.97 |
| $Pop^1$ | 0.99 | 0.96 | 1.00 | 0.97 | 0.85 | 0.98 | 0.84 | 0.80 | 0.79 | 0.92 | 0.66 | 0.98 | 0.98 | 0.92 |
| $Pop^2$ | 0.97 | 0.97 | 0.98 | 0.81 | 0.82 | 0.88 | 0.86 | 0.86 | 0.80 | 0.96 | 0.85 | 0.96 | 0.86 | 0.97 |
| $Lit^2Pop^1$ | 0.99 | 0.99 | 0.99 | 0.89 | 0.90 | 0.86 | 0.92 | 0.90 | 0.87 | 0.94 | 0.78 | 0.99 | 0.98 | 0.99 |
| $Lit^3Pop^1$ | 0.99 | 0.99 | 0.99 | 0.86 | 0.89 | 0.84 | 0.93 | 0.89 | 0.88 | 0.95 | 0.79 | 0.99 | 0.98 | 0.98 |

**Table A2a: Comparison of $\rho$ for ten exponent combinations and 14 countries: Australia (AUS), Brazil (BRA), Canada (CAN), Switzerland (CHE), China (CHN), Germany (DEU), France (FRA), Indonesia (IDN), India (IND), Japan (JPN), Mexico (MEX), Turkey (TUR), United States of America (USA), and South Africa (ZAF). Best fit would mean $\rho=1$. Linear correlation is statistically significant with a p-value lower than 0.05 for all shown countries and combinations.**

| $\beta$ | AUS | BRA | CAN | CHE | CHN | DEU | FRA | IDN | IND | JPN | MEX | TUR | USA | ZAF |
|---|---|---|---|---|---|---|---|---|---|---|---|---|---|---|
| $Lit^1Pop^1$ | 1.02 | 0.79 | 1.10 | 1.07 | 1.05 | 1.01 | 0.96 | 1.21 | 0.96 | 1.28 | 0.76 | 1.49 | 1.01 | 1.07 |
| $Lit^1$ | 0.82 | 0.55 | 0.90 | 0.67 | 0.93 | 0.89 | 0.22 | 0.76 | 0.84 | 0.33 | 0.22 | 0.17 | 0.57 | 0.54 |
| $Lit^2$ | 0.82 | 0.61 | 0.96 | 0.77 | 1.03 | 0.89 | 0.32 | 0.83 | 0.82 | 0.52 | 0.32 | 0.26 | 0.62 | 0.87 |
| $Lit^3$ | 0.82 | 0.63 | 0.99 | 0.84 | 1.06 | 0.88 | 0.38 | 0.86 | 0.82 | 0.64 | 0.38 | 0.32 | 0.65 | 1.05 |
| $Lit^4$ | 0.82 | 0.64 | 1.01 | 0.89 | 1.07 | 0.86 | 0.41 | 0.88 | 0.81 | 0.73 | 0.42 | 0.36 | 0.66 | 1.17 |
| $Lit^5$ | 0.82 | 0.64 | 1.02 | 0.93 | 1.07 | 0.85 | 0.44 | 0.90 | 0.81 | 0.80 | 0.45 | 0.39 | 0.67 | 1.26 |
| $Pop^1$ | 1.01 | 0.66 | 1.01 | 0.87 | 0.68 | 0.92 | 0.47 | 0.77 | 0.84 | 0.66 | 0.55 | 0.61 | 0.88 | 0.65 |
| $Pop^2$ | 1.23 | 0.97 | 1.21 | 1.16 | 0.82 | 1.01 | 2.42 | 1.40 | 1.19 | 1.91 | 1.26 | 2.37 | 1.52 | 1.40 |
| $Lit^2Pop^1$ | 1.03 | 0.81 | 1.12 | 1.12 | 1.04 | 0.99 | 1.09 | 1.26 | 1.01 | 1.45 | 0.82 | 1.70 | 1.04 | 1.22 |
| $Lit^3Pop^1$ | 1.03 | 0.82 | 1.13 | 1.15 | 1.03 | 0.96 | 1.16 | 1.29 | 1.04 | 1.55 | 0.86 | 1.80 | 1.06 | 1.32 |

**Table A2b: Comparison of $\beta$ for ten exponent combinations and 14 countries: Australia (AUS), Brazil (BRA), Canada (CAN), Switzerland (CHE), China (CHN), Germany (DEU), France (FRA), Indonesia (IDN), India (IND), Japan (JPN), Mexico (MEX), Turkey (TUR), United States of America (USA), and South Africa (ZAF). Best fit would mean ($\beta=1$). Linear correlation is statistically significant with a p-value lower than 0.05 for all shown countries and combinations.**

| RMSF | AUS | BRA | CAN | CHE | CHN | DEU | FRA | IDN | IND | JPN | MEX | TUR | USA | ZAF |
|---|---|---|---|---|---|---|---|---|---|---|---|---|---|---|
| $Lit^1Pop^1$ | 1.31 | 1.54 | 1.80 | 2.70 | 1.37 | 1.44 | 1.93 | 5.30 | 2.61 | 2.86 | 1.55 | 3.76 | 1.37 | 1.11 |
| $Lit^1$ | 1.28 | 1.93 | 1.69 | 1.74 | 1.50 | 1.44 | 1.89 | 2.00 | 2.10 | 1.52 | 1.93 | 2.03 | 1.94 | 1.58 |
| $Lit^2$ | 1.28 | 1.83 | 1.51 | 1.85 | 1.42 | 1.36 | 1.68 | 1.86 | 2.03 | 1.41 | 1.77 | 1.77 | 1.81 | 1.37 |
| $Lit^3$ | 1.32 | 1.80 | 1.48 | 2.18 | 1.40 | 1.38 | 1.63 | 1.81 | 2.02 | 1.47 | 1.69 | 1.68 | 1.77 | 1.27 |
| $Lit^4$ | 1.34 | 1.79 | 1.49 | 2.65 | 1.40 | 1.40 | 1.63 | 1.79 | 2.04 | 1.58 | 1.64 | 1.64 | 1.77 | 1.24 |
| $Lit^5$ | 1.37 | 1.78 | 1.53 | 3.25 | 1.40 | 1.42 | 1.66 | 1.79 | 2.06 | 1.70 | 1.60 | 1.63 | 1.77 | 1.23 |
| $Pop^1$ | 1.27 | 1.72 | 1.29 | 1.36 | 1.48 | 1.32 | 1.38 | 2.04 | 1.73 | 1.21 | 1.69 | 1.59 | 1.32 | 1.28 |
| $Pop^2$ | 1.67 | 1.73 | 3.50 | 3.18 | 1.61 | 1.64 | 4.73 | 5.93 | 4.01 | 5.34 | 1.81 | 22.4 | 2.12 | 1.54 |
| $Lit^2Pop^1$ | 1.37 | 1.53 | 2.07 | 4.18 | 1.40 | 1.60 | 2.41 | 6.80 | 3.00 | 4.16 | 1.53 | 6.36 | 1.44 | 1.22 |
| $Lit^3Pop^1$ | 1.41 | 1.53 | 2.27 | 5.74 | 1.41 | 1.69 | 2.75 | 7.64 | 3.23 | 5.29 | 1.52 | 8.31 | 1.50 | 1.32 |

**Table A2c: Comparison of RMSF for ten exponent combinations and 14 countries: Australia (AUS), Brazil (BRA), Canada (CAN), Switzerland (CHE), China (CHN), Germany (DEU), France (FRA), Indonesia (IDN), India (IND), Japan (JPN), Mexico (MEX), Turkey (TUR), United States of America (USA), and South Africa (ZAF). Best fit would mean RMSF=1.**


| | $\rho$ | | $\beta$ | | RMSF | |
|---|---|---|---|---|---|---|
| | Median | IQR | Median | IQR | Median | IQR |
| $Lit^1Pop^1$ | 0.94 | 0.09 | 1.03 | 0.12 | 1.67 | 1.29 |
| $Lit^1$ | 0.81 | 0.29 | 0.62 | 0.44 | 1.82 | 0.40 |
| $Lit^2$ | 0.85 | 0.21 | 0.80 | 0.32 | 1.72 | 0.41 |
| $Lit^3$ | 0.87 | 0.16 | 0.82 | 0.24 | 1.65 | 0.38 |
| $Lit^4$ | 0.89 | 0.14 | 0.82 | 0.24 | 1.64 | 0.36 |
| $Lit^5$ | 0.89 | 0.13 | 0.82 | 0.27 | 1.65 | 0.33 |
| $Pop^1$ | 0.94 | 0.14 | 0.73 | 0.22 | 1.37 | 0.37 |
| $Pop^2$ | 0.87 | 0.12 | 1.24 | 0.33 | 2.65 | 2.87 |
| $Lit^2Pop^1$ | 0.93 | 0.09 | 1.07 | 0.18 | 1.83 | 2.41 |
| $Lit^3Pop^1$ | 0.94 | 0.10 | 1.10 | 0.23 | 1.98 | 3.27 |

**Table A3: Comparison of three skill metrics measuring the fit between modelled and reference nGRP. The table shows the median and IQR over 14 countries computed from the data in Tables A2a-c. Perfect fit would mean a value of one for each metric.**

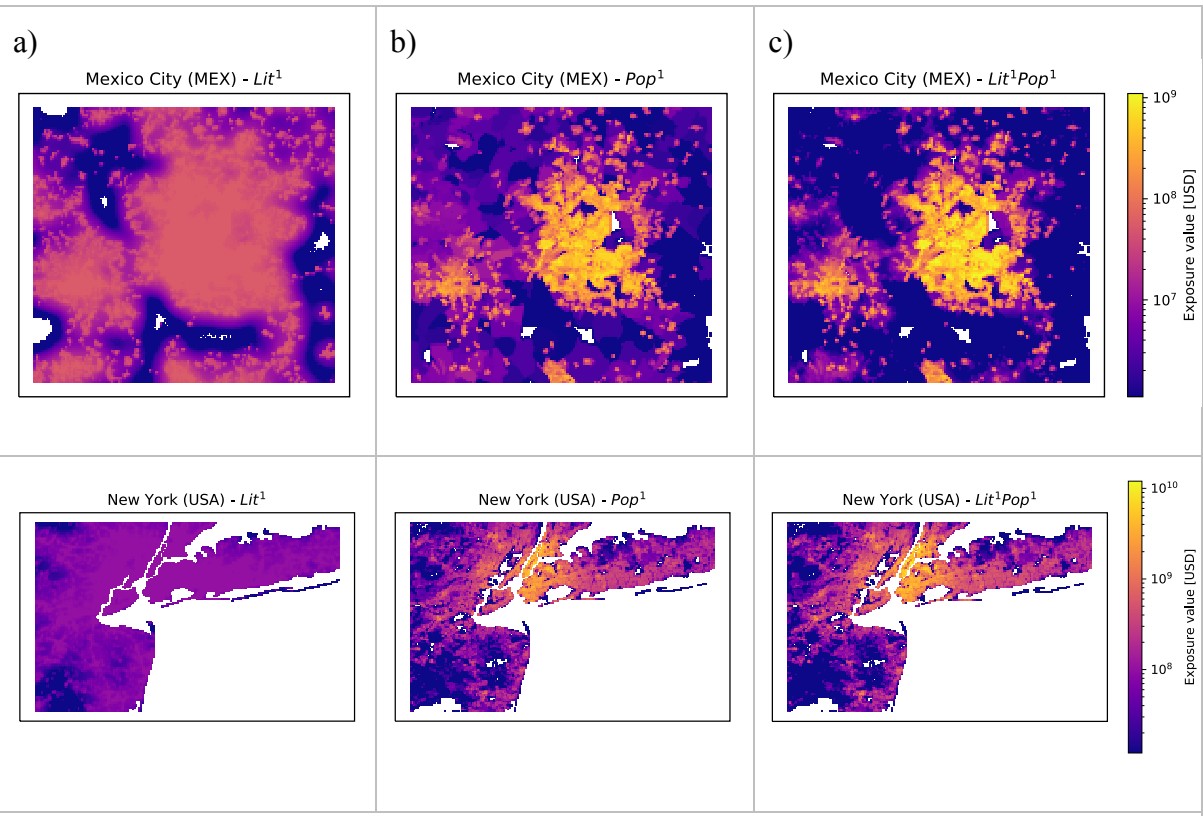

**Figure A1: Maps of disaggregated asset exposure value.** Values are spatially distributed proportional to nightlight intensity of 2016 (*Lit¹*, a), population count as of 2015 (*Pop¹*, b), and the product of both (*Lit¹Pop¹*, c) for Mexico City (MEX) and New York (USA). The maps are restricted to the wider metropolitan areas of Mexico City (99.8-98.6°W; 18.9-20°N) and New York (74.6- 73°W; 40- 41°N) respectively. The colorbar shows asset exposure values in current USD for 2014.

## Author contributions

DS, SE, and DNB developed the method collaboratively. The programming code was written by DS, TR, and SE. Validation and visualization was done by TR and SE. SE prepared the manuscript with contributions from all co-authors.

## Competing interests

The authors declare that they have no conflict of interest.

## Acknowledgements

We would like to thank: Lea Müller for her initial implementation of gridded nightlight as a proxy for global asset exposure, Gabriela Aznar-Siguan for her input regarding the platform CLIMADA and all members of the Weather and Climate Risks Group at ETH Zurich for their inputs and discussions shaping this publication.

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
