# Peer review of "Asset exposure data for global physical risk assessment"

_Earth System Science Data, 2019_

## Referee Comment (RC1) · Anonymous Referee #1 · 14 Nov 2019

Review ESSD-2019-189, global exposure data

Data easy to find and easy to use. Very good metadata, clear licensing. The authors have done a good job of making their codes accessible as well as easy to understand and use.

The authors have not, and can not, validate LitPop to exposure. They have no actual global physical asset data to validate against, at least not at higher spatial resolution than country? They make an interesting case of relevance to GDP (or - if data available - to GRP) but not a definitive case. If they have done careful work to produce a trustworthy product - as I believe they have - perhaps they need substantial caution in extrapolating consistency and predictive skill to global scales. The tool sort of works for 14 rich countries for GDP. As exposure examples they show only 4 cities. Extrapolation to a global exposure data product remains very much a work in progress. The skillful combination of night light data and population data, as reproduced here, represents a useful but still small step? This statement (from the discussion section, line 305): "top-down approach implemented here does not account for differences in infrastructure types and vulnerability" seems to this reviewer to represent a more accurate and honest statement than their expansive title. Recommend publication after a better statement of actual accomplishment / progress.

Specific comments:

Line 46: "With global satellite images being publicly available and updated regularly, it has been proven to be an useful source". Awkward. Authors intend singular 'it' to refer to nightlight data but in this sentence they confuse readers by the plural reference to satellite 'images'. They could clarify by writing 'global nightlight images' …, 'they' have proven …? Need some change to smooth this out.

Line 55: Reader will find no Zhao et al. 2017 reference. Later (line 141) reader encounters "Naizhuo Zhao et al., 2017" with a matching citation in the reference list. Please fix one or the other and then use consistently? Again at line 161. Please check throughout the manuscript, you do not want to get this particular reference wrong.

Line 85: NASA produces the VIRS nightlight product used here but technically the data come from the Suomi NPP's Visible Infrared Imaging Radiometer Suite where NPP indicates a joint NASA NOAA effort. Other ESSD papers that reference nightlight data (for emissions purposes) use the NOAA DMSP URL rather than the NASA VIRS link promoted here? E.g. https://ngdc.noaa.gov/eog/ dmsp/download_radcal.html. Some remote sensing papers compare VIRS to DMSP, favor of VIRS, but gridded emissions products tend to use DMSP? Emissions products tend to want fires but this population product tends to avoid fires? Here (line 89) these authors use the term 'stable lights' but most readers will not understand that term as excluding fires? For remote sensing community, some clarification useful here?

Line 87: Better to use ISO units for times, e.g. 0130 and 1330?

Line 104: "selected for this application, because unlike other spatial population datasets, it does not incorporate" Change punctuation here to: '… selected for this application because, unlike other spatial population datasets, it does not incorporate …'

Line 108: "both spatial and temporal resolution" You mean temporal overlap or time step coincidence, rather than resolution? Resolution would suggest, annual, monthly, etc., when in fact you have used only 2012 and 2016 for nightlight while GPW has 2010 and 2015? (On line 115 you refer to time steps rather than temporal resolution.)

Line 118, 119: "no direct damage to the value of the land itself in the case of disaster" Authors need to justify this default assumption. For coastal land masses subject to wind, water current and sea level/inundation damage, land values almost certainly change pre- to post-disaster, sometimes extensively. For example, termination or increased cost of flood zone insurance, as does and even more should happen post-storm, changes land values? Local governments and commercial real estate firms notorious for artificially maintaining land-values at pre-storm levels to thereby maintain tax bases and market values? Hurricane loss and damage community publishes many assessments on land values before and after storm landfall?

Line. 120: A substantial literature exists on weakness of national GDP reports as indicators of economic output. Perhaps not relevant here? If relevant, authors need to justify why they use GDP?

Line 134: "wide range of income groups and world regions". But, OECD data already filter out a large number of countries/economies? Therefore one might gain a wide range of OECD data, but not actually a wide range of global data? The list of 14 countries presented here looks more like G-7 plus BRIC, e.g. not exactly a wide range of global economies or regions? In line 135 the authors admit "bias towards developed and emerging economies". "wide range" is not correct.

Line 234: What is "$Pop^2$"? Earlier we have seen and understood $Lit^m Pop^n$ with m and n as weighting factors. In Figure 3 and Table A3 the reader now encounters Lit with values 1 (default) through 5, Pop with values 1 and 2, and LitPop with m of 2 and 3. In plain terms, we see examples with Lit weighted normally to heavily, Pop weighted normally to some increased value, and the LitPop combination with Lit at weights of 3 and 4. One can tease out the meanings and processes but one does not know the weighting factors? Weight of m = 2 means double? 20%? 2 orders of magnitude? It will follow that m = 3 indicates thrice? 30%? three orders of magnitude? From equation (1) m and n look like exponents, so $Lit^5$ indicates Lit to the 5th, e.g. 5 orders of magnitude? Mathematically correct, one suspects, but meaning obscure. Why did authors choose to vary Lit more than Pop? Given the strong valid preference for LitPop (with m = n = 1) what does a reader learn by seeing all these permutations? In line 279 the authors use the word "multiplicative" but, for this reader, that term differs substantially from exponential? Later (line 290) the authors use the word "exponent".

Again, one assumes they know what they did, but they have not conveyed their approach clearly to this reader.

Line 240: "is the most  adequate combination"

Line 243: "In the validation in Section 3.2, Mexico shows" Because the authors do not mention Mexico in Section 3.2, this sentence should instead read 'Compared to' or 'In contrast to'?

Line 245: "the smaller districts" You mean smaller economically, not smaller geographically?

Line 253: "housing and infrastructure in suburban México that is used by a population that works in the city". This rural or suburban pattern of residence coupled with employment/work in a central district must represent a very common or even predominant pattern in most large cities? E.g. Rio, Jakarta, New York, even Mumbai? Not clear to this reviewer why Mexico City would represent an outlier in this regard?

Lines 268, 269: "performs well across countries from different continents and income groups" I already questioned this supposed broad coverage (see comment for line 134, above) and authors in their text have admitted that this is not true. Given the OECD filters and limited availability of data, the authors should show much more caution with broad statements like this?

Line 268: "Global" consistency. Authors presented data from 14 highly-selected countries. This subsample hardly qualifies as global. We do not even know - at least from this paper - what percentage of global population, global nightlights, or global GDP their subset represents. Substantial, perhaps (at least in 2012 and 2016), but hardly definitive?

Line 274: "income group 1" Does this text refer to a World Bank or IPCC categorization? Reader has not encountered group numbers? In lines 136, 137 authors referred to lower-middle-income and low income groups. 'group 1' refers to these income levels? Reader must seek out table legend for Table A1 to learn with group 1 means.

Line 294: "the LitPop exposure model" by this point in this manuscript, this reader views this phrase with deep dis-satisfaction and suspicion. The authors showed possibly valid (but highly geographically limited) LitPop to GDP correlations but they have in no way advanced to a LitPop to exposure model. As they say themselves!

Lines 311 to 319, openness replicability etc. Excellent section! Could / should prove useful example for other ESSD papers.

---

## Referee Comment (RC2) · Anonymous Referee #2 · 26 Nov 2019

The authors present an open-source method that can be used to downscale low-resolution economic predictors to high-resolution gridded data by using nightlight intensity and gridded population data. The method and required data are described and a validation of the methodology is conducted for 14 selected countries. A global high-resolution dataset for 227 countries is created using this method and openly available for download. The documentation of the method in the open-source archive CLIMADA and the dataset are state-of-the-art and easily assessable to users. The presentation of the method and the dataset within the present manuscript needs major improvements. In general, the method and the dataset are described incompletely, the validation exercise and the subsequent consequences appear ad-hoc and unmotivated. In particular, the manuscript lacks a clear and precise writing style in various locations that make it

difficult for the reader to follow. Important information is missing, appears in different locations or is poorly referenced. This is a data description paper so all the relevant information concerning the data (including the input data) should be assembled here. Besides the specific locations noted below I ask the authors to critically revise the full manuscript to improve readability and understanding. I had to (re-)read many parts of the manuscript several times to finally get the full picture.

Major points:

1. The manuscript is about a global exposure dataset (for asset and/or GDP exposure?) for 227 countries. However, most of the manuscript deals with validation of the 14 test countries and some metropolitan areas. I expect the authors to include a description of the full dataset in section 3. This should include a clear statement of the countries and time periods included, missing countries or regions with low coverage in the available dataset and maybe even a worldmap figure. The reader should not download the huge dataset or consult Worldbank data in order to obtain this information himself.

2. The name of the method 'LitPop' and the function 'LitPop' (sometimes in italic) are used as synonyms. This is VERY confusing for the reader. To avoid confusion I would strongly encourage the authors to use LitnPopm (with m and n in the exponent) every time you talk about the function, even in the case when the exponent is one you should write Lit1Pop1 (with ones in the exponent).

3. Although I am not an expert on nightlight data I have the impression that there are some subtleties involved the user should know about. I quick google search tells me that usually an exponent >1 for nightlight data is used when deriving economic proxies to partially deal with the saturation issue. (This is somehow also apparent from your results in Figure 3). What about latitude-dependence of light intensity and the influence on your global dataset? I think the discussion in section 2 on input data needs to be advanced so the reader really gets to know the dataset and its subtleties.

[Figure]

4. In line 144 the authors state: 'While the absolute value of LitPop in itself does not bear any interpretable meaning, its relative value in comparison to the national or subnational sum determines how much of a macroeconomic indicator each pixel receives'. Saying that the authors do exactly the opposite in their validation exercise: they use aggregated absolute values from their method to compare it to observed quantities. Even more, this point is very difficult to extract from the manuscript. Only after jumping back and forth in the manuscript I understood that they actually calibrate their method with national GDP data and then compare subnational estimates. This needs to be stated much clearer.

5. The functionality used in Eq. 1 seems rather ad-hoc and only motivated by a study used in China. Have the authors used different approaches, different functional dependencies? What were their findings? Why is the exponential scaling beneficial? Mathematically, only the relative weighting between Lit and Pop is changed by the two exponents. Therefore, the approach could be simplified by using only one exponent that reflects the relative difference between both contributions. Have the authors looked into this direction?

6. The authors say they use two skill scores in line 178. Later they widen their analysis to three skill scores (e.g. Fig 3), which they interchangeably call methods as well. The authors should adjust their manuscript accordingly and stick to one naming convention.

7. The relevance of skill score 'beta' remains obscure (line 185). First, it is fully unclear about what slope the authors are talking. Second, the concept of linear regression in this context is fully unclear. Third, skill score 'beta' basically contains the same information as 'rho' (eq. 4), it is just a different scaling with respect to the standard deviations. I therefore do not understand why beta is needed in the first place and would ask the authors to remove one of them (beta or rho) as they are based on the same information.

8. The range of exponents m and n explored in the validation seems random and bares

any motivation. What is the motivation for the range of exponents explored? I strongly encourage the authors to motivate their validation and to conduct a more stringent validation accordingly.

9. The validation section (3.2) needs to clarified and amended. On first read (see my point raised before) I had the understanding that the analysis around Fig. 3 is based on 14 data points only (E.g. the caption of Fig. 3 points at this as well). Only later I understood that the authors use many data points (14 x subnational regions). The number of data points is never mentioned, however. How is the interquartile range defined? Please be much more precise and proactive.

10. Based on the redundancy of either beta or rho (stated above) and your diverging findings for different skill scores within your validation, I find the final decision to use the downscaling with m=n=1 (line 240) very ill-founded. At this stage I would expect a more thorough and stringent assessment of the different exponents and functionalities (see comment above). Otherwise, the full validation exercise seems redundant.

11. Line 187 (and others in the following): the notion of economically strong (or large) and weak regions is not very well defined. The reader can sort of understand what the authors hint at but it remains very unclear. How do they distinguish strong from weak regions? What is the precise criterion? Does this hold nationally or internationally?

12. The sentence 'There is probably a lot of housing and infrastructure in suburban México that is used by a population that works in the city and thus contributes to the GRP of Mexico City' (strange comparison of stocks and flows) and the following discussion is very difficult to digest for the non-expert reader. I find this discussion very relevant and think it should be extended here or at some other point in the manuscript as it directly links to many relevant issues: a) What does nightlight intensity actually capture? Assets or GDP? b) What is the highest downscaling resolution one should aim at when population is most likely a better proxy of the location of assets but nightlight also captures economic activity (e.g. driving cars)? Also in the light of above

sentence when GDP and assets seem to be separated by municipal boundaries. c) how can the interaction of both data sources most efficiently be combined? How does the present methodology add to this discussion? How can the different exponents be interpreted in this respect?

13. The discussion in line 284-292 is very vague as it is very hard to judge for the reader when to apply the authors' recommendation: high-resolution vs. coarsely resolved?, use a higher exponent of nightlights instead. . . instead to what? Why use exponent n=3 when this was never a potentially recommended value in the validation before? The discussion on auxiliary data should be placed somewhere else.

14. It is very unfortunate that the validation was (or could) only be conducted for 14 countries and no low-income country. The subsequent application of this method to all countries globally has to be treated with caution. In the present manuscript I am missing a detailed discussion of the reliability of the dataset for specific regions and/or income groups and a discussion of potential workarounds. What is the result of the authors' validation in terms of income groups? Is there any information (e.g. trends with income) that could be valuable for low income countries not treated here? What about very small countries, islands, etc? How could other data sources (e.g. household survey data from the Worldbank) be used to improve the data? What has been conducted with this respect in the literature so far (c.f. following paper and the references cited there: Gunasekera, R., et al. (2015). "Developing an adaptive global exposure model to support the generation of country disaster risk profiles." Earth-Science Reviews 150: 594-608.)?

15. The concept of intermediate downscaling appears in line 257 very ad-hoc and is used thereafter without further explanation.

16. LitPop as a top-down approach is first introduced in line 302. It would make much more sense to make this statement much earlier otherwise one should avoid this notion in general.

17. The term 'exposure' is used differently throughout the manuscript. It seems that the authors use it for 'asset exposure' but this is not fully clear. Exposure is very general and could be understood as population or GDP exposure as well. Therefore, I encourage the authors to be more precise and use the expression 'asset exposure' every time they mean it.

18. All abbreviations (e.g. GDP, GRP), all variables, and all subscripts (e.g. pix) need to be explained at first use, even if the authors think that they are self-explanatory. Thereafter another redefinition should be avoided and the authors should stick to their abbreviations.

19. Figure 4: The usage of Mexico (country) and México (region) is very confusing for the reader. Clearly state this difference and maybe use 'México region' to underline the difference.

Minor points:

20. The discussion in line 31 should include another freely available gridded GDP dataset: Kummu, M., et al. (2018). "Gridded global datasets for Gross Domestic Product and Human Development Index over 1990-2015." Sci Data 5: 180004.

21. The reference Murakami et al is outdated. Please update to: Murakami, D. and Y. Yamagata (2019). "Estimation of Gridded Population and GDP Scenarios with Spatially Explicit Statistical Downscaling." Sustainability 11(7).

22. Line 34: The statement on high-resolution GDP data availability for academic purposes only is not true. Upon checking the reference I found that the data is freely available. The corresponding reference should be included in the manuscript: Geiger, Tobias; Daisuke, Murakami; Frieler, Katja; Yamagata, Yoshiki (2017): Spatially-explicit Gross Cell Product (GCP) time series: past observations (1850-2000) harmonized with future projections according to the Shared Socioeconomic Pathways (2010-2100). GFZ Data Services. http://doi.org/10.5880/pik.2017.007

23. Line 55 (and others): the reference to Zhao et al. cannot be found in the list of references.

24. Line 177: What does the exponent '5' stand for in nGRP_i? Looks like a footnote which I am unable to locate. Same issue in line 189 and 228.

25. Line 182: Seems like the separated equation for rho got lost and appears inline now. The enumeration eq. 4 is also missing.

26. Figure 2: I do not understand what do you mean by log-normal colorbar? I would appreciate the colorbar to have a label. What kind of USD do you use here? PPP-adjusted, current or real? This applies similarly for Fig A1.

27. Line 219: replace top -> bottom

28. Line 326-328: The information on RMSF is repeating what the authors mentioned earlier around line 190.

29. Line 240: remove 'an'

30. Line 243: A reference to the data in the appendix would be very helpful here as the reader is unable to extract the information for Mexico from section 3.2.

31. Line 264: the reference for Pittore et al cannot be found in the list of references.

32. Line 334: replace get > become

33. Caption figure A1: replace 'the Mexico and USA' > 'Mexico and the USA'

---

## Author Comment (AC1) · 10 Feb 2020

***ESSD-2019-189, Exposure data for global physical risk assessment***

***Author comment***

*We would like to thank the two anonymous referees for their thorough review and the supportive comments, questions and recommendations regarding the manuscript.*

*On the following pages, please find the comments of both reviewers, the author responses and revisions to the text as realised in the revised manuscript.*

***Responses to RC1 by Anonymous Referee #1:***

1.0.

Data easy to find and easy to use. Very good metadata, clear licensing. The authors have done a good job of making their codes accessible as well as easy to understand and use.

The authors have not, and can not, validate LitPop to exposure. They have no actual global physical asset data to validate against, at least not at higher spatial resolution than country? They make an interesting case of relevance to GDP (or - if data available - to GRP) but not a definitive case. If they have done careful work to produce a trustworthy product - as I believe they have - perhaps they need substantial caution in extrapolating consistency and predictive skill to global scales. The tool sort of works for 14 rich countries for GDP. As exposure examples they show only 4 cities. Extrapolation to a global exposure data product remains very much a work in progress. The skillful combination of night light data and population data, as reproduced here, represents a useful but still small step? This statement (from the discussion section, line 305): "top-down approach implemented here does not account for differences in infrastructure types and vulnerability" seems to this reviewer to represent a more accurate and honest statement than their expansive title. Recommend publication after a better statement of actual accomplishment / progress.

Response: We would like to thank the anonymous reviewer for the positive comments as well as for the questions and suggestions for improvements of the paper. In the following, we would like to respond to the single aspects mentioned above, like the issue of validation and the need for a clearer communication of the actual accomplishments and limitations of the generic methodology for global asset exposure disaggregation presented in our publication.

The objection to the claim for "validation" is well conceived. We agree that the term suggests a more direct evaluation of asset value disaggregation than what we can provide. What we are in fact doing, is to use the related socio-economic flow variable GDP for an evaluation of the LitPop disaggregation approach, comparing a variety of exponent combinations, i.e. changing $m$ and $n$ in $Lit^m Pop^n$ (see also our detailed reply to comments 1.10 and 2.14).

Following both reviewers' concerns regarding the claim of validation, we will rename "validation" to "evaluation" and revise the manuscript in various places in order to

communicate and discuss the limitations and the purpose of the evaluation in a more accurate way.

Besides the lack of rigorous validation for stated reasons, the evaluation provides confidence that the disaggregation of national asset values proportionally both to nightlight intensity (Lit) and population count (Pop) enables us to provide a first-order estimation of gridded global asset exposure that mitigates some limitations of using Lit or Pop alone (i.e. blooming, saturation and lack of resolution as discussed in the paper).

Already in the first submitted version of the manuscript, it is stated that the LitPop method was not evaluated for developing countries, and not evaluated against physical asset values but with GDP alone. In line with reviewers' requests, we have strengthened these messages in the revised paper, including parts of the Discussion, too. Additionally, we did strengthen our call for validation against local empirical data to increase confidence, especially before using the data set in developing countries. The specific changes in the discussion can be found in the responses to the specific comments below (c.f. comments 1.15-1.17)

We did indeed pick up the recommendation for a better statement of actual accomplishment and limitations. We have more precisely stated the limitations in the Discussion, but did also include a clearer statement on usability to avoid misunderstandings. In the following, we give an overview of the most relevant changes in response to this. Please note that this general comment also informs changes in other part of the text, that were further revised in response to specific comments of both reviewers, and to the second reviewer's call for mature revisions to improve readability (comment 2.0). Revisions to the text are thus reported in response to the specific comments.

- Revisions in the abstract with regards to the validation/evaluation:

Old: *"To evaluate the predictive skill of the downscaling approach, GDP distributed proportional to LitPop to subnational administrative regions is compared to reference values. The results for 14 countries show that the predictive skill of LitPop is higher than using nightlights or population data alone."*

Revised: *"Due to the lack of reported subnational asset data, the disaggregation methodology cannot be validated for asset values. Therefore, we compare disaggregated GDP per subnational administrative region to reported gross regional product (GRP) values for evaluation. The comparison for 14 industrialized and new-industrialized countries shows that the disaggregation skill for GDP using nightlights or population data alone is not as high as using a combination of both data types."*

- Renaming of subsections related to the validation/ evaluation as follows: *"2.7 Validation of the Downscaling"* → *"2.6 Evaluation"*; *"3.2 Validation"* → *"3.2 Evaluation"*.
- Additionally, the term of "validation" is replaced by "evaluation" in other parts of the manuscript and further clarifications are added. For instance, the following explanation is added in the Data and Methods Section, in the subsection now named *"2.6 Evaluation"*: *"The LitPop approach's skill in disaggregating asset exposure cannot be evaluated directly due to the lack of reference asset value data on a subnational level. Therefore, GDP and GRP are used instead for an indirect evaluation of the methodology. GDP and GRP are used to assess the subnational*

> *disaggregation skill, comparing varying combinations of the exponents m and n in Lit$^{m}$Pop$^{n}$."*

- *… and further discussed in the Discussion and the Conclusion of the revised manuscript:*
  Discussion: *"For the gridded exposure dataset presented here, the LitPop methodology is used to disaggregate total asset values. Due to a lack of subnational reference asset values, the LitPop methodology's performance for the downscaling of asset stock values could not be evaluated directly. The assessment of disaggregation skill was instead based on the flow variables GDP and GRP. Given a correlation between stocks and flows within each country, this approach represents an indirect evaluation of the methodology for asset exposure downscaling."* (c.f. comment 1.8)
  Conclusion: *"However, the methodology could not be evaluated directly against subnational asset data and the evaluation based on GDP was limited to 14 OECD countries. Therefore, the asset exposure data is not suitable for applications with a local or sector-specific focus without further validation."*

*Specific comments:*

1.1. Line 46: "With global satellite images being publicly available and updated regularly, it has been proven to be an useful source". Awkward. Authors intend singular 'it' to refer to nightlight data but in this sentence they confuse readers by the plural reference to satellite 'images'. They could clarify by writing 'global nightlight images' ..., 'they' have proven ...? Need some change to smooth this out.

Response: We revised the sentence according to the reviewer's suggestion (new version in *blue*):

*"Being publicly available and updated regularly, global nightlight images have been proven to be a useful source […]"*

1.2. Line 55: Reader will find no Zhao et al. 2017 reference. Later (line 141) reader encounters "Naizhuo Zhao et al., 2017" with a matching citation in the reference list. Please fix one or the other and then use consistently? Again at line 161. Please check throughout the manuscript, you do not want to get this particular reference wrong.

Response: We have corrected for the faulty reference. The particular publication is now referenced as *Zhao et al. (2017)* throughout the manuscript.

1.3. Line 85: NASA produces the VIRS nightlight product used here but technically the data come from the Suomi NPP's Visible Infrared Imaging Radiometer Suite where NPP indicates a joint NASA NOAA effort. Other ESSD papers that reference nightlight data (for emissions purposes) use the NOAA DMSP URL rather than the NASA VIRS link promoted here? E.g. https://ngdc.noaa.gov/eog/ dmsp/download_radcal.html. Some remote sensing papers compare VIRS to DMSP, favor of VIRS, but gridded emissions products tend to use DMSP? Emissions products tend to want fires but this population product tends to avoid fires? Here (line 89) these authors use the term 'stable lights' but most readers will

not understand that term as excluding fires? For remote sensing community, some clarification useful here?

Response: We would like to thank for the recommended changes in referencing the Black Marble nightlight product. The nightlights data used for this data set and within all nightlight based modules within the CLIMADA modelling framework is the NASA Earth Observatory as referenced in the manuscript (https://earthobservatory.nasa.gov/features/NightLights/page3.php). To ensure reproducibility, we need to keep the reference to the actual download of the data accurate. The NOAA DMSP URL provided by the reviewer links to an DMSP product and other VIIRS that are not necessarily identical to the Black Marble tiles provided by NASA and used here. According to a study published by Munich Personal RePEc Archive in 2019, "VIIRS night lights data are a better proxy for economic activity than are the more widely used DMSP data." (Gibson, John and Olivia, Susan and Boe-Gibson, Geua (2019): *Which Night Lights Data Should we Use in Economics, and Where?, https://mpra.ub.uni-muenchen.de/97582/*). The main reasons being that VIIRS signal has less noise and is better suited to pick up dim light sources than DMSP.

In order to clarify the reference, we added the citation, now also referring to the accompanying publication: *Román et al. (2018)*: "NASA's Black Marble nighttime lights product suite" [https://www.sciencedirect.com/science/article/pii/S003442571830110X]. This agrees with Aznar-Siguan and Bresch (2019) who use the same nightlight product (https://www.geosci-model-dev.net/12/3085/2019/)

Temporary fires are not relevant for our use of the nightlight data. They are not part of the stable lights (as clarified below). Sustained and spatially fixed fires, i.e. from industrial burning processes, are likely to contribute to the nightlight intensity in the Black Marble product. This agrees with the purpose of using nightlights as a proxy for human economic activity.

To clarify the term "stable lights", we now also refer to Román et al. (2018) in Section 2.2:

*"To isolate luminosity from sustained human activity, the Black Marble nightlight product includes corrections for Lunar artefacts, cloud, terrain, atmospheric, snow, airglow, stray light, and seasonal effects (Carlowicz, 2017; Lee et al., 2014; Román et al., 2018)*

(C.f. response to comment 2.3 for for further elaborations on the nightlight data used).

1.4. Line 87: Better to use ISO units for times, e.g. 0130 and 1330?

Response: The notation with "am" and "pm" is indeed not ISO. Thus, we changed the time format in Section 2.2 to *"01:30"* and *"13:30"* according to ISO 8601. ISO 8601 formats time of the day as follows: hh:mm:ss. Reference: https://www.iso.org/iso-8601-date-and-time-format.html (fee required for access) or for a free overview: https://en.wikipedia.org/wiki/ISO_8601.

1.5. Line 104: "selected for this application, because unlike other spatial population datasets, it does not incorporate" Change punctuation here to: '... selected for this application because, unlike other spatial population datasets, it does not incorporate ...'

Response: The punctuation in the cited sentence in Section 2.3 was corrected and "this application" replaced by "the LitPop methodology" to communicate more clearly:

*"[…] selected for the LitPop methodology because, unlike other spatial population datasets, […]"*

1.6. Line 108: "both spatial and temporal resolution" You mean temporal overlap or time step coincidence, rather than resolution? Resolution would suggest, annual, monthly, etc., when in fact you have used only 2012 and 2016 for nightlight while GPW has 2010 and 2015? (On line 115 you refer to time steps rather than temporal resolution.)

Response: According to the reviewer's suggestion, we changed the text in Section 2.3 according as follows:

Old: *"[…], in terms of both spatial and temporal resolution."*

Revised: *"[…], both in terms of spatial resolution and available time steps."*

1.7. Line 118, 119: "no direct damage to the value of the land itself in the case of disaster" Authors need to justify this default assumption. For coastal land masses subject to wind, water current and sea level/inundation damage, land values almost certainly change pre- to post-disaster, sometimes extensively. For example, termination or increased cost of flood zone insurance, as does and even more should happen post- storm, changes land values? Local governments and commercial real estate firms notorious for artificially maintaining land-values at pre-storm levels to thereby maintain tax bases and market values? Hurricane loss and damage community publishes many assessments on land values before and after storm landfall?

Response: We agree with the reviewer, that there are scenarios in which natural hazard affects the value of land. At the same time, the data set focuses on tangible / physical assets. Just as cultural or emotional values exposed to disasters are not considered, we also exclude the value of the land itself. Since the mark-up of 24% is a simple multiplicative factor in the produced capital accounts, land value can be included by users of the data set by multiplying all values with the factor 1.24. For clarification, we added the following two sentences in Section 2.4.1:

*"While not universally true, this assumption is based on the focus of the asset exposure data for the purpose of assessing direct impact to tangible structures. For applications considering the impact on the value of land, the linear scale-up can be reapplied before utilization of the asset exposure data."*

1.8. Line. 120: A substantial literature exists on weakness of national GDP reports as indicators of economic output. Perhaps not relevant here? If relevant, authors need to justify why they use GDP?

Response: Thank you for pointing out the wider discussion on the limitations of GDP. The main reason to use GDP for evaluation is that national GDP data is available globally and sub-national GDP (i.e. GRP) is available for a variety of countries. The reason for this is that GDP is a standard that is used widely in research and outside academia, and thus GDP numbers are provided by many governmental and international agencies. Discussing the

pros and cons of using GDP is not the scope of this paper, as it is mainly used for the evaluation of the disaggregation.

In the revised manuscript, we state the limitations that arise by evaluating the downscaling approach with an economic flow variable but applying it to a stock variable in the Discussion of the revised manuscript:

*"For the gridded exposure dataset presented here, the LitPop methodology is used to disaggregate total asset values. Due to a lack of subnational reference asset values, the LitPop methodology's performance for the downscaling of asset stock values could not be evaluated directly. The assessment of disaggregation skill was instead based on the flow variables GDP and GRP. Given a correlation between stocks and flows within each country, this approach represents an indirect evaluation of the methodology for asset exposure downscaling."*

1.9. Line 134: "wide range of income groups and world regions". But, OECD data already filter out a large number of countries/economies? Therefore one might gain a wide range of OECD data, but not actually a wide range of global data? The list of 14 countries presented here looks more like G-7 plus BRIC, e.g. not exactly a wide range of global economies or regions? In line 135 the authors admit "bias towards developed and emerging economies". "wide range" is not correct.

Response: This criticism of the term "wide range" is well taken, as discussed already in response to the general comment 1.0. To clarify this limitation of the selected countries, added the information to the abstract that the 14 countries are all industrialized or newly-industrialized. For further transparency, we changed the sentences in Section 2.4.3 of the revised manuscript as follows (changes in *blue*):

*"The aim of the selection was to include countries from  as wide as possible of income groups and world regions. Since the selection of countries was limited by the availability of GRP data, 135 the selection has a bias towards  industrialized and newly industrialized OECD member states. According to World Bank income groups, these countries include eight countries from the high-income group (World Bank income group 4), four countries from the upper-middle-income group (3), two countries from the lower-middle-income (2), and no countries from the low-income group (1)."*

1.10. Line 234: What is "$Pop^2$"? Earlier we have seen and understood $Lit^m Pop^n$ with m and n as weighting factors. In Figure 3 and Table A3 the reader now encounters Lit with values 1 (default) through 5, Pop with values 1 and 2, and LitPop with m of 2 and 3. In plain terms, we see examples with Lit weighted normally to heavily, Pop weighted normally to some increased value, and the LitPop combination with Lit at weights of 3 and 4. One can tease out the meanings and processes but one does not know the weighting factors? Weight of m = 2 means double? 20%? 2 orders of magnitude? It will follow that m = 3 indicates thrice? 30%? three orders of magnitude? From equation (1) m and n look like exponents, so $Lit^5$ indicates Lit to the 5th, e.g. 5 orders of magnitude? Mathematically correct, one suspects, but meaning obscure. Why did authors choose to vary Lit more than Pop? Given the strong valid preference for LitPop (with m = n = 1) what does a reader learn by seeing all these permutations? In line 279 the authors use the word

"multiplicative" but, for this reader, that term differs substantially from exponential? Later (line 290) the authors use the word "exponent". Again, one assumes they know what they did, but they have not conveyed their approach clearly to this reader.

Response: Thank you again for pointing out that there can be confusion about what *m and n* are in the Method and the Results sections of the paper. As stated in line 150 (first submission) in the Methods section 2.5, m and n are the exponents of Lit and Pop respectively. I.e. $Lit^2Pop^1 = Lit*Lit*Pop$ at each grid cell. That's why the combination could also be called multiplicative (c.f. line 279). A high exponent for *Lit* (f.i. $Lit^4$) does not only mean that *Lit* is weighted more against *Pop* for each pixel, but also that bright pixels (i.e. with large values of *Lit*) are weighted even higher against dim pixels with low values of *Lit* (within the same country).

For a more detailed discussion regarding the exponents, please also refer to our response to comment 2.5.

We will take up this comment by calling m and n 'exponents' more consequently throughout the text. In Section 2.5 of the revised manuscript, we additionally add the following explanation:

*"Changing the exponents m and n determines with which power the two input variables contribute to the disaggregation function. The exponents m and n do not only weight relatively between Lit and Pop but they also determine the contrast in the distribution between all grid cells within a country. The larger the exponent, the more value is concentrated on grid cells with large values of Lit or Pop respectively. The aim of the evaluation described in Section 2.6 is to compare disaggregation skill of varied combinations of m and n and select the most adequate combinations for subnational disaggregation."*

Ad *"Why did authors choose to vary Lit more than Pop?":* The exponent combinations chosen in this publication were derived based on literature and iteration: In previous studies, exposure has been estimated by disaggregation either proportionally to $Pop^1$ (e.g. Gunasekera et al., 2015) or $Lit^m$ with m>1 to account for the exponential relationship between nightlight intensity and economic indicators as discussed by Zhao et al. (2017) among others (e.g. Aznar-Siguan and Bresch, 2019). Based on this, we varied exponents but stopped when the skill scores are expected to get only worse or stagnate. This is possible because of a monotonous effect of changing the exponent. For instance, from $Pop^1$ to $Pop^2$, all three skill scores perform worse. (c.f. $Pop^2$ in Figure 3a-c).. Considering the drop in performance of $Pop^2$ compared to $Pop^1$ with regards to these skill indicators, there is no point in considering higher exponents (i.e. $Pop^3$, $Pop^4$). A larger exponent is expected to lead to an even lower performance.

Ad *"Given the strong valid preference for LitPop (with m = n = 1) what does a reader learn by seeing all these permutations?":* We had no a priori preference for *m = n = 1*. The different combinations of the two exponents were analyzed to find the best performing combination of Lit and Pop. Thus, showing the combinatiions displays the results justifying our choice of $Lit^1 * Pop^1$ over other combinations of *Lit* and *Pop*.

Addressing the comments of both anonymous reviewers concerning the clarity of the disaggregation function, we have rewritten Sections 2.5 / 2.6. In this process, we combined both sections into one section (2.5).

In this process, the mentioning of "*m = n = 1*" as a default was deleted from section 2.6, since suggesting this combination as a "default" is only derived further down and it is rather a choice than a default.

Additionally, we have revised the opening sentences of Section 3.2, providing more details on the evaluation process. The paragraph now reads as follows:

*"To evaluate the performance of the LitPop methodology, we compute and compare the disaggregation skill in regards to GDP for varying exponents m and n in $Lit^mPop^n$ (Eq. 1 and 2). Here, we show the comparison based on 14 countries with a total of 507 regional GRP data points available and ten combinations of m and n: $Lit^1Pop^1$, $Lit^1$, $Lit^2$, $Lit^3$, $Lit^4$, $Lit^5$, $Pop^1$, $Pop^2$, $Lit^2Pop^1$, and $Lit^3Pop^1$. These exponent combinations were selected based on examples in the literature and then explored iteratively, stopping at combinations with decreased skill compared to lower order combinations. The 14 countries make up 67% (USD 168 trillion) of the total dataset's exposure and 64.5% (USD 52 trillion) of global GDP in 2014. For each country and exponent combination, the median and the spread of three skill metrics are compared: $\rho$, $\beta$, and RMSF (Fig. 3 and Tables A2 and A3)."*

1.11. Line 240: "is the most an adequate combination"

Response: We corrected the typo in Section 3.2, removing "an".

Revision: *"is the most  adequate combination"*

1.12. Line 243: "In the validation in Section 3.2, Mexico shows" Because the authors do not mention Mexico in Section 3.2, this sentence should instead read 'Compared to' or 'In contrast to'?

Response: The reference to Section 3.2 is indeed not precise. The low correlation coefficients are not shown directly in Section 3.2 but in Table A2a in the Appendix (column labeled MEX). Please note that we have rearranged the results section in response to the second reviewer's call for more clarification. We have adjusted the text to clarify the reference. New opening paragraph in the revised manuscript (including changes in response to other comments):

*"The skill metrics for the subnational disaggregation of GDP in the country Mexico shows low value of $\rho$ compared to most other countries for all tested values of m and n ($\rho$=0.76 for $Lit^1Pop^1$, c.f. Table A2a). The example of Mexico is presented here to illustrate limitations and uncertainties of the disaggregation approach. Figure 5 shows the data behind the evaluation for Mexico, i.e. modelled and reference nGRP for all 32 districts of Mexico. The corresponding plot data can be found in Table S2 as supplementary material. While the LitPop methodology performs well for most of the districts with relatively low GRP, it fails to reproduce reference nGRP for the main (capital) metropolitan region consisting of the districts México and Mexico City (Distrito Federal)."*

1.13. Line 245: "the smaller districts" You mean smaller economically, not smaller geographically?

Response: We would like to thank the reviewer for pointing out this ambiguity. Please refer to our response to comment 2.11 by the second reviewer for a more detailed discussion regarding this point.

Revision: we have specified the statement as follows (changes in *blue*):

*"the  districts with relatively low GRP"*

1.14. Line 253: "housing and infrastructure in suburban México that is used by a population that works in the city". This rural or suburban pattern of residence coupled with employment/work in a central district must represent a very common or even predominant pattern in most large cities? E.g. Rio, Jakarta, New York, even Mumbai? Not clear to this reviewer why Mexico City would represent an outlier in this regard?

Response: We agree that this phenomenon is not specific to Mexico City. We chose to show Mexico because the split of the whole Mexico City area into two administrative regions allows us to illustrate this phenomenon with our analysis.

Please note that we have revised the whole section on the example of Mexico in response to comments 1.12-1.14 and especially comment 2.12 by the second reviewer. Please refer to our response to comments 1.12 and 2.12 for a more detailed discussion and a revised version of the Section.

1.15. Lines 268, 269: "performs well across countries from different continents and income groups" I already questioned this supposed broad coverage (see comment for line 134, above) and authors in their text have admitted that this is not true. Given the OECD filters and limited availability of data, the authors should show much more caution with broad statements like this?

Response: As discussed in the response to the reviewer's comment 1.0, we are following the rightfully cautioning remarks of the reviewer. Therefore, we have reformulated the claims of performance in Section 4 (Discussion) of the revised manuscript as follows:

Old: *"It should be noted that due to lack of data we were not able to evaluate the method's performance for low income countries (World Bank income group 1). Therefore, the application of the asset exposure data for local assessments in countries within this income group should be treated with caution. Another caveat to global consistency is the fact that the quality and resolution of the underlying population dataset varies between countries, as discussed in greater detail in the next paragraph. As a consequence of these limitations, asset exposure data should be validated against local data before application for local risk assessments, especially in low income countries."*

Revised: *"However, the evaluation of the of the methodology's disaggregation skill presented here is limited to an assessment of disaggregation skill for 14 OECD countries. It should be noted that due to lack of data we were not able to evaluate the method's performance for low income countries (World Bank income group 1). Therefore, the application of the asset exposure data for local assessments in countries within this income group should be treated with caution. Another caveat to global consistency is the fact that the quality and resolution of the underlying population dataset varies between countries, as discussed in greater detail in the next paragraph. As a consequence of these limitations,*

*asset exposure data should be validated against local data before application for local risk assessments, especially in low income countries."*

1.16. Line 268: "Global" consistency. Authors presented data from 14 highly-selected countries. This subsample hardly qualifies as global. We do not even know - at least from this paper - what percentage of global population, global nightlights, or global GDP their subset represents. Substantial, perhaps (at least in 2012 and 2016), but hardly definitive?

Response: Thank you for this helpful remark. We call the data set "globally consistent" because consistent data and methods were applied for all countries. We agree that the evaluation does not provide a global validation of the dataset, however, they do represent around one third of the total global asset exposure. In order to better communicate the global consistency of the data set, we thus provide a world map of the exposure data in the beginning of the revised Results section. In addition, we provide numbers of total GDP, and asset values represented for all countries for which exposure data has been made available, as well as for the 14 countries used for evaluation.

In Section 3.1 of the revised manuscript, we now provide a world map (new Figure 2) and the following information:

*"We applied the LitPop methodology with the exponents m = n = 1 to compute gridded asset exposure data for 224 countries and areas worldwide (Fig. 2). Total physical asset values of 2014 were disaggregated proportionally to Lit$^1$Pop$^1$ to a grid with the spatial resolution of 30 arcsec (approximately 1 km). Total asset values in the dataset sum up to $2.51*10^{14}$ (251 trillion) current USD of 2014. The 140 countries with produced capital data used as total asset value, contribute USD 245 trillion (97.6 %) to the total asset exposure. The remaining 84 countries where asset values were estimated from GDP and a GDP-to-wealth ratio instead, contribute the remaining USD 6 trillion. In total, the 224 countries contribute around 99.9% to recorded global GDP."*

In Section 3.2, we add the following information:

*"The 14 countries make up 67% (USD 168 trillion) of the total data set's exposure and 64.5% (USD 52 trillion) of global GDP in 2014."*

A brief discussion of this is added in Section 4 (Discussion):

*"While the presented data set is not complete, it provides data for the countries contributing 99.9% of global GDP."*

1.17. Line 274: "income group 1" Does this text refer to a World Bank or IPCC categorization? Reader has not encountered group numbers? In lines 136, 137 authors referred to lower-middle-income and low income groups. 'group 1' refers to these income levels? Reader must seek out table legend for Table A1 to learn with group 1 means.

Response: We agree that the income group definition should be stated ealier in the manuscript. We thus revise the manuscript to specify the income group definition (World Bank income group) in both paragraphs referred to by the reviewer (c.f. response to comment 1.9), changes in *blue*:

Section 2.4.3 (GRP):

*"According to World Bank income groups, these countries include eight countries from the high-income group (World Bank income group 4), four countries from the upper-middle-income group (3), two countries from the lower-middle-income (2), and no countries from the low-income group (1). The il*ncome groups and data sources per country are listed in Table A1 in the Appendix."*

Section 4 (Discussion):

*"It should be noted that due to lack of data we were not able to evaluate the method's performance for  low income countries (World Bank income group 1).*

1.18. Line 294: "the LitPop exposure model" by this point in this manuscript, this reader views this phrase with deep dis-satisfaction and suspicion. The authors showed possibly valid (but highly geographically limited) LitPop to GDP correlations but they have in no way advanced to a LitPop to exposure model. As they say themselves!

Response: We agree that the terminology used here was misleading. We have changed the wording in the specified line to *"LitPop methodology"*. (C.f. response to comment 2.2. regarding the use of the term LitPop).

1.19. Lines 311 to 319, openness replicability etc. Excellent section! Could / should prove useful example for other ESSD papers.

Response: We would like to thank the reviewer again for pointing this out. This is much appreciated.
* * *
*Review ESSD-2019-189, global exposure data*

**Responses to RC2 by Anonymous Referee #2**

2.0.

The authors present an open-source method that can be used to downscale low-resolution economic predictors to high-resolution gridded data by using nightlight intensity and gridded population data. The method and required data are described and a validation of the methodology is conducted for 14 selected countries. A global high-resolution dataset for 227 countries is created using this method and openly available for download. The documentation of the method in the open-source archive CLIMADA and the dataset are state-of-the-art and easily assessable to users. The presentation of the method and the dataset within the present manuscript needs major improvements. In general, the method and the dataset are described incompletely, the validation exercise and the subsequent consequences appear ad-hoc and unmotivated. In particular, the manuscript lacks a clear and precise writing style in various locations that make it difficult for the reader to follow. Important information is missing, appears in different locations or is poorly referenced. This is a data description paper so all the relevant information concerning the data (including the input data) should be assembled here. Besides the specific locations noted below I ask the authors to critically revise the full manuscript to

improve readability and understanding. I had to (re-)read many parts of the manuscript several times to finally get the full picture.

Response:

We would like to thank the reviewer for the synthesis and the key comment regarding readability, as well as for the detailed suggestions for improvement. In response to the reviewers' general recommendations concerning readability and understanding, as well as the specific comments, we have applied major revisions to the manuscript. They concern the terminology but also the general structure of the Sections Data & Methods and the Results. Most specific changes are found in response to the more specific comments of both reviewers.

For a better reading experience, we introduced a more precise and consistent naming of the methodology as "LitPop" (c.f. comment 2.2). In addition, we are now consequently calling the resulting data "asset exposure data" (c.f. comment 2.17). We also replaced the term "validation" with the more fitting term of "evaluation" (c.f. comment 1.0). The term "disaggregation" is now used more consequently throughout the manuscript to refer for the core process of distributing asset values proportionally the combination of nightlight intensity and population data.

We rearranged the Results section to start with the main result of the publication which is the global asset exposure data set that is available online (Section 3.1). This section now also includes a world map (new Figure 2) to give a quick impression of the data. The plots of disaggregated asset exposure data in metropolitan areas used for a qualitative evaluation of the disaggregation (Figure 4) are moved down below the quantitative evaluation.

We are confident that these and many other revisions in the manuscript included in direct response to the specific comments of the two reviewers help to increase the general readability and strengthen the message of the manuscript, and again want to thank the anonymous reviewers for their useful suggestions.

Please refer to the responses to the more specific comments below for further elaborations and specific revisions.

*Major points:*

2.1. The manuscript is about a global exposure dataset (for asset and/or GDP exposure?) for 227 countries. However, most of the manuscript deals with validation of the 14 test countries and some metropolitan areas. I expect the authors to include a description of the full dataset in section 3. This should include a clear statement of the countries and time periods included, missing countries or regions with low coverage in the available dataset and maybe even a worldmap figure. The reader should not download the huge dataset or consult Worldbank data in order to obtain this information himself.

Response: In response to this very valid request of the reviewer, we added an overview of all used input data including source and reference year in the new **Table 1**. Structured information and meta data on all countries was previously not provided or only as part of the data repository. Detailed information per country is now presented in Table S1 in the Supplementary Materials. This includes metadata for all 224 countries included in the asset exposure data set and 26 countries and areas not included. In addition, more detail on the total asset value data are now given in the text in Section 2.4.1. During the revision

process, we realized that the actual numbers of countries with data provided is 224 and not 227. We would like to apologize for this mistake which is now corrected for in the current version of the manuscript. The three omitted areas that were previously mistakenly included into the list of countries with data are British Indian Ocean Territory, French Southern Territories, and South Georgia.

In response to the comment, we have included more information on data sources and countries in the Sections Data and Methods (Table 1 and Section 2.4.1), Results (Section 3.1), as well in Table S1 in the Supplement.

Addition in Section 2.4.1 of the revised manuscript ("Total asset value per country"):

*"Out of a total of 250 countries we considered for the production of this dataset, produced capital numbers for 2014 are available for 140 countries. For these 140 countries, produced capital for 2014 was used here as total asset value for disaggregation. For additional 87 countries, total asset values were set to non-financial wealth. Non-financial wealth was computed from the country's GDP and the GDP-to-wealth ratio estimates derived from the Credit Suisse Research Institute's Global Wealth Report (Credit Suisse Research Institute, 2017). This approach has previously been followed by Geiger et al. (2018). We compared produced capital and non-financial wealth for 140 countries (Table S1 in the Supplement) and found that non-financial wealth can be used as a conservative approximation of produced capital. For 59 of the 87 countries without produced capital data, an average GDP-to-wealth ratio of 1.247 was applied. In summary, the whole dataset contains gridded asset exposure data for a total of 224 countries, ignoring 26 countries and areas due to lack of data. Missing countries and areas (with currently assigned ISO 3166-1 alpha-3 codes) are Aland Islands, Antarctica, Bonaire, British Indian Ocean Territory, Sint Eustatius and Saba, Bouvet Island, Cocos (Keeling) Islands, Christmas Island, Guadeloupe, French Guiana, French Southern Territories, Heard Island and McDonald Islands, Holy See, Kosovo, Libya, Martinique, Mayotte, Pitcairn, Palestine, Reunion, South Georgia and the South Sandwich Islands, South Sudan, Svalbard and Jan Mayen, Syrian Arab Republic, Tokelau, United States Minor Outlying Islands, and Western Sahara.*

*An overview over the utilized data per country, including, produced capital (were available), GDP-to-wealth ratios, and GDP for 2014 is provided in Table S1."*

Addition to Section 3.1 ("Global gridded asset exposure"): *Please refer to comment 1.16.*

2.2. The name of the method 'LitPop' and the function 'LitPop' (sometimes in italic) are used as synonyms. This is VERY confusing for the reader. To avoid confusion I would strongly encourage the authors to use LitnPopm (with m and n in the exponent) everytime you talk about the function, even in the case when the exponent is one you should write Lit1Pop1 (with ones in the exponent).

Response: We appreciate the reviewer's feedback concerning the ambiguity of the term "LitPop" in the manuscript. We have revised the terminology throughout the manuscript (including figures and tables) in order to make it more precise and easier to follow for the readers. We revised the terminology based on the suggestions by the reviewer. To go into detail, here are the now consistently used terms and variables used in the revised manuscript, including the key sentences and definitions in the revised manuscript:

The term "LitPop" names the asset exposure disaggregation methodology described and evaluated in the manuscript. Thus, "LitPop" spelled that way signifies neither the resulting data set nor a specific function.

The variable "$Lit^m Pop^n$" signifies a gridded digital number that is computed per pixel according to Equation 1 (Section 2.5). Briefly said, "$Lit^m Pop^n$" is computed per pixel by multiplying nightlight intensity to the power of m with population count to the power of n.

The LitPop methodology is not limited to the realization with $Lit^1 Pop^1$. To make this difference clear, the exponents m and n are now always written, even when they are equal to one. Only in cases where an exponent is set to zero, the entire part is omitted, i.e. $Lit^2$ instead of $Lit^2 Pop^0$.

Changes in key paragraphs in the manuscript (revised text in *blue*):

- LitPop:

*"Here, we are using and expanding the "lit population" approach presented by Zhao et al. (2017) to define and implement a globally consistent methodology for asset exposure disaggregation, named LitPop hereafter."* (Introduction)

- *$Lit^m Pop^n$*:

*"In a first step, the two gridded input datasets are interpolated linearly to the same resolution of 30 arcsec, or coarser resolution if desired. Then, the combination of the two aforementioned datasets is conducted for each pixel grid cell:*

$$Lit^n Pop^m{}_{pix} = \left(NL_{pix} + \delta\right)^n \cdot Pop_{pix}{}^m \qquad\qquad (1)$$

*Where the LitPop digital number value $Lit^m Pop^n{}_{pix}$ per grid cell (*pix*) is computed from the grid cell's nightlight intensity $NL_{pix} \in [0, 255]$, and population count $Pop_{pix} \in \mathbb{R}^+$, as well as the exponents $n, m \in \mathbb{N}$. For all m > 0, the added $\delta$ is equal to 1 to ensure that non-illuminated but populated pixels grid cells do not get assigned zero value. In the case that nightlight data is used on its own without population data (m = 0), $\delta$ is set to 0."* (Section 2.5 in Data and Methods)

We have also adjusted the wording in the schematic overview of the methodology (Figure 1), to agree with the revised terminology and moved Figure 1 down to Section 2.5 to make it easier to refer to the figure when reading the technical description of the methodology.

2.3. Although I am not an expert on nightlight data I have the impression that there are some subtleties involved the user should know about. I quick google search tells me that usually an exponent >1 for nightlight data is used when deriving economic proxies to partially deal with the saturation issue. (This is somehow also apparent from your results in Figure 3). What about latitude-dependence of light intensity and the influence on your global dataset? I think the discussion in section 2 on input data needs to be advanced so the reader really gets to know the dataset and its subtleties.

Response:

*Regarding the exponent >1 usually used for nightlight data:*

Thanks for pointing out that nightlight intensity is usually taken to a power larger than 1 for deriving economic proxies (corresponding to $Lit^2$, $Lit^3$, $Lit^4$, and $Lit^5$ in the manuscript) The

exponential relationship between nightlight intensity and economic activity as it was also pointed out by Zhao et al. (2017) upon which the LitPop methodology is based. As we also discuss in the manuscript with reference to the combination of nightlight intensity with population data is an alternative approach to mitigate the limitations arising from the saturation effect of nightlight data, among others. Indeed, Zhao et al. (2017) discuss empirical evidence for an exponential relationship between nightlight intensity and population density. This relationship makes it more tangible why $Lit^m Pop^n$ has been considered as an alternative downscaling function in the first place. For the evaluation, varying combinations of the exponents m and n are compared precisely for the purpose to decide which exponential combination to use for the production of a global asset exposure data set and to discuss the best alternatives (Sections 2.6 and 3.2 in the revised manuscript). As the reviewer has noted rightly, the disaggregation of GDP to subnational units based on *Lit* alone performs indeed better for m>1 than for m=1. However, in combination with *Pop*, larger exponents m>1 lead to a decrease in skill.

Changes to the manuscript: We have revised the Introduction in response to several comments by the two anonymous reviewers, both to increase readability and include information and references that had been missing before. Among these changes is a more detailed discussion of the use of nightlight data and the approach presented by Zhao et al. (2017). The most crucial revised paragraph regarding above comment is the following:

*"As a consequence of saturation, socioeconomic indicators scale rather exponentially than linearly with nightlight intensity (Zhao et al., 2015, 2017). To counteract the saturation effect, Aznar-Siguan and Bresch (2019) used squared nightlight intensity as a basis for asset exposure disaggregation. Saturation and blooming can also be mitigated by combining nightlights with other data types: Zhao et al. (2017) enhanced nightlight intensity values with population data to get a more accurate estimation of spatial economic activity in China. This is based on the observation that there is also an exponential relationship between nightlight intensity and population density. The authors showed that the product of nightlight intensity and gridded population count (called "lit population" by the authors), is a better indicator for economic activity in China than nightlight intensity alone."*

*Regarding latitude dependencies in nightlight data:*

A potential source of latitude dependency in nightlight data are so called stray lights (Lee et al., 2014: "The S-NPP VIIRS Day-Night Band On-Orbit Calibration/Characterization and Current State of SDR Products"). This effect is affecting the high latitudes in both hemispheres. The VIIRS Day-Night Band includes an automated stray light correction (Roman et al., 2018 and Lee et al., 2014). No other relevant latitude dependency issues concerning VIIRS products are known to us. If there were other large scale geographical biases in the data or the relationship between nightlight intensity and asset values, they would be partly mitigated by the fact that the LitPop methodology is applied country by country. This means that gridded $Lit^m Pop^n$ is normalized per country before disaggregation. Thus, inconsistencies between countries are irrelevant.

In order to inform the readers on corrections applied to the nightlight intensity product used here, we add the following sentence in Section 2.2 of the revised manuscript:

*"To isolate luminosity from sustained human activity, the Black Marble nightlight product includes corrections for Lunar artefacts, cloud, terrain, atmospheric, snow, airglow, stray light, and seasonal effects (Carlowicz, 2017; Lee et al., 2014; Román et al., 2018)."*

2.4. In line 144 the authors state: 'While the absolute value of LitPop in itself does not bear any interpretable meaning, its relative value in comparison to the national or subnational sum determines how much of a macroeconomic indicator each pixel receives'. Saying that the authors do exactly the opposite in their validation exercise: they use aggregated absolute values from their method to compare it to observed quantities. Even more, this point is very difficult to extract from the manuscript. Only after jumping back and forth in the manuscript I understood that they actually calibrate their method with national GDP data and then compare subnational estimates. This needs to be stated much clearer.

Response: Once again we would like to thank the reviewer to point out an unclear formulation in the manuscript. We agree that the term "interpretable meaning" is misleading here and that further revisions are required to explain the downscaling approach more clearly.

The actual value of $Lit^mPop^n$ is a unitless digital number per grid cell. A meaningful gridded data set is produced when normalized $Lit^mPop^n$ is multiplied by a country's total GDP or asset value. As formulated by Zhao et al. (2017): *"Lit population does not correspond to any measurement unit in real life, representing neither people count nor brightness of nighttime lights. It indicates economically weighed-population"*

Changes in the manuscript: In response to the comment, we have refined the text in several places to communicate the following points more clearly, most prominently by providing a revised brief overview of the methodology (Section 2.1), also focusing on the fact that we use total indicator values (i.e. total asset value or GDP) per country and disaggregate them proportional to (normalized) gridded $Lit^mPop^n$. Revised version:

**"2.1 Overview**

*The core functionality of the LitPop methodology is to spatially disaggregate national total asset values to obtain a gridded asset exposure product. Gridded nightlight intensity (Section 2.2) and gridded population data (Section 2.3) are combined to compute a digital number at grid cell level (Section 2.5). Physical asset stock values (i.e. produced capital, Section 2.4.1) are then disaggregated proportional to the digital number per grid cell (Section 2.5). This results in the gridded asset exposure dataset presented here. Table 1 provides a detailed overview over the input data.*

*Instead of the physical asset stock, GDP (Section 2.4.2) or gross regional product (GRP, Section 2.4.3) can be distributed to obtain GDP per grid cell. Because of a lack of subnational produced capital data, GDP and GRP are used to evaluate the methodology by assessing the subnational disaggregation skill for varied combinations of the input data, as described in Section 2.6."*

Additional explanation added in Section 2.5 of the revised manuscript before Equation 2:

*"In a second step, gridded $Lit^mPop^n$ is taken as a relative representation of economic stocks at each grid cell. It is used to linearly disaggregate a total asset values of a country to a geographical grid. More precisely, the value of $Lit^mPop^n_{pix}$ relative to the sum of*

*Lit^mPop^n over all pixels within the boundaries of the country determines how much of a total value is assigned to each grid cell: […]"* (Section 2.5 in Data and Methods)

2.5. The functionality used in Eq. 1 seems rather ad-hoc and only motivated by a study used in China. Have the authors used different approaches, different functional dependencies? What were their findings? Why is the exponential scaling beneficial? Mathematically, only the relative weighting between Lit and Pop is changed by the two exponents. Therefore, the approach could be simplified by using only one exponent that reflects the relative difference between both contributions. Have the authors looked into this direction?

This comment spans different related points and questions regarding the disaggregation methodology presented and evaluated in the paper. We are responding to the more general questions on the selection of the functionality first and treat the specific question concerning the exponents and weighting between *Lit* and *Pop* separately further below.

Ad *"The functionality used in Eq. 1 seems rather ad-hoc and only motivated by a study used in China. Have the authors used different approaches, different functional dependencies? What were their findings?":*

Disaggregating national aggregated socioeconomic indicators proportional to gridded nightlight or population data is a common approach. The decision to work with these two input data types is based (i) on the requirement of a methodology that is based on data that is globally – and freely – available, as well as easily reproducible and updatable; and (2) on the analysis by Zhao et al. (2015, 2017) showing the benefits of combining the two datasets for an admittedly regional study (yet of considerable spatial extent). As explained by Zhao et al. as well as in the Introduction of our manuscript, the combination is mitigating the saturation and blooming artefacts in the nightlight data and exploiting the exponential relationships both between nightlight intensity and economic activity on the one hand, and between nightlight intensity and population density on the other hand. Furthermore, nightlights and population data can be seen as partly complementary representations of asset distribution patterns: Patterns of residential assets correspond well to population while high nightlight intensity correlate well with commercial and industrial assets (i.e. Gunasekera et al., 2015, reference suggested by this reviewer and gratefully acknowledged and integrated into the revised version of the manuscript, c.f. comment 2.14). According to Gunasekera et al. (2015), "infrastructure assets follow a spatial pattern similar to population distribution" (with particular patterns for specific sectors like power production), and literature shows a strong link between "relationship between area of night-time lights and GDP economic activity" and that there is a "correlation of high artificial light intensity at night with urban non-residential areas such as commercial and industrial." Furthermore, risk assessment studies have used higher exponents of nightlight intensity (corresponding to *Lit^m*) to mitigate the saturation effect (e.g. Aznar-Siguan and Bresch, 2019; Gettelman et al., 2017; Sutton and Costanza, 2002). Based on these previous works, we are confident to combine *Lit* and *Pop* in the applied functional form to disaggregate asset value. The limitations of this approach are discussed in detail in the revised manuscript, for instance in the first three paragraphs of the Discussion (Section 4). The exponents *m* and *n* in Equation 1 can be varied to obtain different multiplicative combinations of the two input data sets (i.e. *Lit^2Pop^1 = Lit * Lit ***

*Pop* and *Pop$^2$ = Pop * Pop*). We have used this in the evaluation for a comparison of disaggregation proportional to nightlight intensity or population data alone with combined data, also for different exponents. We have not conducted any comparisons with disaggregation functionalities based on additional data, as we aimed at a globally applicable methodology. The scope of this paper is not to attempt to fit a predictive model (e.g. in the form of I = a * Lit$^b$ + c * Pop$^d$ or other functional forms…) for the countries with GRP data available. The reason being that for a single country, we could fit a function if data is available, but this is not available on a global level and our main requirement for the methodology was global consistency. That's why we decided to provide data based on a global disaggregation proportional to *Lit$^m$Pop$^n$* with the same exponents *m* and *n* used for all countries. The evaluation was undertaken to assess the quality and applicability of the disaggregation approach at all and to suggest the most appropriate combination of *m* and *n* based on the data available on a globally consistent basis. Nevertheless, for a specific single country or regional use, any user *might* choose to fit *m* and *n* to any (locally) available data (as the Python code does allow to do so easily).

Changes to the revised manuscript: We have revised the background of the LitPop methodology in the Introduction, better explaining the approach by Zhao et al. (2017) and including more literature making use of non-linearly scaled nightlight data (e.g. Aznar-Siguan and Bresch, 2019; Gettelman et al., 2017; Sutton and Costanza, 2002) and the combination of nightlight intensity with population data for socioeconomic disaggregation (Sutton et al, 2007).

Revision in the Introduction (changes in *blue*):

*"Brightness can exude from bright pixels to neighboring pixels, causing the brightness in the latter to be overestimated, leading to blooming. This issue occurs especially in large urban areas and on specific surfaces, such as sand and water (Elvidge et al., 2004; Small et al., 2005). As a consequence of saturation, socioeconomic indicators scale rather exponentially than linearly with nightlight intensity (Sutton and Costanza, 2002; Zhao et al., 2015, 2017). To counteract the saturation effect, Gettelman (2017) and Aznar-Siguan and Bresch (2019) used exponentially scaled nightlight intensity as a basis for GDP disaggregation for tropical cyclone risk assessments.*  *Saturation and blooming can* also *be mitigated by combining nightlights with other data types: Sutton et al. (2007) combined the areal extend of lit area with population data to estimate GDP at a subnational level. Zhao et al. (2017) enhanced nightlight intensity values with population data to get a more accurate estimation of spatial economic activity in China. This is based on the observation that there is also an exponential relationship between nightlight intensity and population density.*  *The authors showed that*  *the product of nightlight intensity and gridded population count (called "lit population" by the authors), is a better*  *proxy for economic activity in China than nightlight intensity alone."*

*Ad "Why is the exponential scaling beneficial? Mathematically, only the relative weighting between Lit and Pop is changed by the two exponents. Have the authors looked into this direction?":*

There is most likely a misunderstanding concerning Equation 1: The exponents m and n do not only weight relatively between *Lit* and *Pop* but they also determine the contrast in the distribution between all grid cells within a country. The larger the exponent, the more value

is concentrated on grid cells with large values of *Lit* or *Pop* respectively. For illustration, we have drafted an example with a toy country consisting of 6 toy grid cells:

**A) Absolute values**

| Grid cell ID | *Lit* | *Pop* | $Lit^2$ | $Lit^1Pop^1$ | $Lit^2Pop^2$ | $Lit^2Pop^1$ | $Lit^4Pop^2$ |
|---|---|---|---|---|---|---|---|
| 1 | 5 | 500 | 25 | 2500 | 6E+06 | 12500 | 2E+08 |
| 2 | 5 | 10000 | 25 | 50000 | 3E+09 | 250000 | 6E+10 |
| 3 | 50 | 500 | 2500 | 25000 | 6E+08 | 1E+06 | 2E+12 |
| 4 | 50 | 10000 | 2500 | 500000 | 3E+11 | 3E+07 | 6E+14 |
| 5 | 200 | 500 | 40000 | 100000 | 1E+10 | 2E+07 | 4E+14 |
| 6 | 200 | 10000 | 40000 | 2E+06 | 4E+12 | 4E+08 | 2E+17 |

**B) Normalized values**

| Grid cell ID | *Lit* | *Pop* | $Lit^2$ | $Lit^1Pop^1$ | $Lit^2Pop^2$ | $Lit^2Pop^1$ | $Lit^4Pop^2$ |
|---|---|---|---|---|---|---|---|
| 1 | 0.01 | 0.02 | 0.00 | 0.00 | 0.00 | 0.00 | 0.00 |
| 2 | 0.01 | 0.32 | 0.00 | 0.02 | 0.00 | 0.00 | 0.00 |
| 3 | 0.10 | 0.02 | 0.03 | 0.01 | 0.00 | 0.00 | 0.00 |
| 4 | 0.10 | 0.32 | 0.03 | 0.19 | 0.06 | 0.06 | 0.00 |
| 5 | 0.39 | 0.02 | 0.47 | 0.04 | 0.00 | 0.04 | 0.00 |
| 6 | 0.39 | 0.32 | 0.47 | 0.75 | 0.94 | 0.90 | 0.99 |

The grid cells all have values of Lit of 5, 50, or 200 and values of Pop of 500 or 10000, in permuted combinations. The top table shows the absolute values for varied exponents in $Lit^mPop^n$, the bottom tables shows normalized values for the same data (i.e. each column has the sum 1.0). Please note the differences between $Lit^1Pop^1$ and $Lit^2Pop^2$ and between $Lit^2Pop^1$ and $Lit^4Pop^2$ respectively: The larger the exponents, the more *normalized* value is concentrated at grid cell 6 with the largest values of *Lit* and *Pop*. Thus, the approach could not be reproduced using only one exponent as suggested by the reviewer.

For clarification, we included the following sentences into the revised script at the end of Section 2.5:
*"The exponents m and n do not only weight relatively between Lit and Pop but they also determine the contrast in the distribution between all grid cells within a country. The larger the exponent, the more value is concentrated on grid cells with large values of Lit or Pop respectively."*

2.6. The authors say they use two skill scores in line 178. Later they widen their analysis to three skill scores (e.g. Fig 3), which they interchangeably call methods as well. The authors should adjust their manuscript accordingly and stick to one naming convention.

Response: In order to improve the readability, we have revised the wording in the Methods and the Results sections: We now consistently refer to three "skill metrics", namely $\rho$, $\beta$, and RMSF and do not refer to them as "scores" or "methods" anymore.

2.7. The relevance of skill score 'beta' remains obscure (line 185). First, it is fully unclear about what slope the authors are talking. Second, the concept of linear regression in this context is fully unclear. Third, skill score 'beta' basically contains the same information as 'rho' (eq. 4), it is just a different scaling with respect to the standard deviations. I therefore do not understand why beta is needed in the first place and would ask the authors to remove one of them (beta or rho) as they are based on the same information.

Response: In the evaluation, we are using the Pearson correlation coefficient $\rho$ and the slope of linear regression $\beta$ as complementary skill metrics. The metrics are computed from modelled (i.e. disaggregated) vs. reference (i.e. reported) normalized gross regional product (nGRP) per subnational unit (region, district, etc.) within each country. The confusion around the linear regression used in the evaluation is well taken. We agree that $\rho$ and $\beta$ are not completely independent metrics.

Still, the metrics $\rho$ and $\beta$ do indeed convey complementary information on the disaggregation skill of the applied combinations of exponents in the disaggregation function: $\rho$ tells us to which degree modelled and reported nGRP correlate, i.e. if there is a strong linear relationship between the two variables. However, there could be a perfect correlation ($\rho$=1) with a very steep or very flat slope $\beta$. For this reason, $\rho$ is sometimes referred to as a measure of "potential skill". The slope $\beta$ tells us, whether there is a systematic over- or underestimation of economically large regions in the disaggregated data. For $\beta$>1, economically large regions are overestimated (as we found for $Pop^2$), for $\beta$<1, economically large regions are underestimated by the disaggregation (as we found for $Lit^1$). However, $\beta$ alone does not convey any information on the correlation between the two variables, i.e. there could be a perfect slope ($\beta$=1) for variables with a very weak linear relationship.

In more mathematical terms: $\beta$ is calculated by scaling ratio of the stanrd deviations of the two variables with $\rho$ ($\beta = \rho \cdot \sigma_{mod} / \sigma_{ref}$). While $\sigma_{ref}$ is constant for each country, $\sigma_{mod}$ changes with changing exponents $m$ and $n$. Thus, $\beta$ is indeed not just a scaled version of $\rho$. Instead, it carries additional information as discussed above. We could also use $\rho$ and $\sigma_{mod} / \sigma_{ref}$ instead of $\rho$ and $\beta$ to assess the disaggregation skill. The reason we use $\beta$ is that by scaling the standard deviations with $\rho$, the slope has a more straight-forward meaning related to the aim to select a combination of m and n that produces the best linear relationship between modelled and reference nGRP. Based on our arguments presented above, we propose to keep considering both metrics for evaluation.

For clarification, we include parts of above reasoning in Section 2.6 of the revised manuscript (changes in *blue*):

"*The correlation coefficient $\rho$ is a widely used  metric and straight forward to interpret and communicate: A value of 1  indicates a perfect positive linear correlation between the two variables while a value of 0  indicates that there is no linear correlation. However, $\rho$ is no direct measure of the deviations of nGRP$_{mod}$ from nGRP$_{ref}$ and yields no information regarding the slope of the linear relationship. Therefore, it only represents a potential skill and needs to be evaluated in combination with a measure of the*

*slope. The slope of the linear regression* conveys the information, whether there is a systematic over- or underestimation of economically large regions in the disaggregated data. Therefore, $\beta = \rho \cdot \sigma_{mod}/\sigma_{ref}$ *is calculated to complement the analysis:* $\beta$ *larger (lower) than 1 implies an overestimation (underestimation) of the GRP of*  regions with relatively large GRP *and an underestimation (overestimation) of*  regions with relatively low GRP *in the downscaling. Together,* $\rho$ *and* $\beta$ *allow for an evaluation of the linear fit between modelled and reference data.*"

2.8. The range of exponents m and n explored in the validation seems random and bares any motivation. What is the motivation for the range of exponents explored? I strongly encourage the authors to motivate their validation and to conduct a more stringent validation accordingly.

Response: Please refer to the response to comment asked in comment 1.10: *"Why did authors choose to vary Lit more than Pop?"*

In Section 3.2 of the revised manuscript, we add the following clarifying statement:

*"These exponent combinations were selected based on examples in the literature and then explored iteratively, stopping at combinations with decreased skill compared to lower order combinations."*

2.9. The validation section (3.2) needs to clarified and amended. On first read (see my point raised before) I had the understanding that the analysis around Fig. 3 is based on 14 data points only (E.g. the caption of Fig. 3 points at this as well). Only later I understood that the authors use many data points (14 x subnational regions). The number of data points is never mentioned, however. How is the interquartile range defined? Please be much more precise and proactive.

Response: We would like to thank both reviewers for their various comments regarding the validation section in the manuscript. Based on this and other comments, we have thoroughly revised both sections on that topic (both in the Data and Methods and in the Results). Most prominently, we have replaced the term "validation" with "evaluation" (c.f. response to comment 1.0).

In response to the particular misunderstanding brought up here regarding the exact methodology applied for the evaluation and the number of data points used, we revised the text both in Section 3.2 and 2.6 (formerly 2.7), as well as in the newly introduces data overview in Table 1. The aim of the revision is to point out more clearly now that the evaluation is based on 507 data points, corresponding to normalized reported and modelled GRP data for a total of 507 sub-national regions in 14 countries.

We added the following clarifying statement in Section 2.6 in Data and Methods:

*"The LitPop approach's skill in disaggregating asset exposure cannot be assessed directly due to the lack of reference asset value data on a subnational level. Therefore, GDP and GRP are used instead for an indirect evaluation of the methodology. GDP and GRP are used to assess the subnational disaggregation skill, comparing varying combinations of the exponents m and n in $Lit^m Pop^n$."*

In Section 3.2 (Results) of the revised manuscript, we replaced the opening paragraph as follows:

Old: *"The downscaling within countries is validated by comparing the downscaled and reported subnational GDP with three 225 quantitative methods. The Pearson correlation coefficient $\rho$, linear slope parameter $\beta$, and root-mean-squared fraction RMSF per country are shown in Tables A2. To compare the overall performance of the different methods, median and spread of the scores are compared in Figure 3 and Table A3."*

Revised: *"To evaluate the performance of the LitPop methodology, we compute and compare the disaggregation skill in regards to GDP for varying exponents m and n in $Lit^mPop^n$ (Eq. 1 and 2). Here, we show the comparison based on 14 countries with a total of 507 regional GRP data points available. The 14 countries make up 67% (USD 168 trillion) of the total dataset's exposure and 64.5% (USD 52 trillion) of global GDP in 2014. Ten combinations of m and n are assessed: $Lit^1Pop^1$, $Lit^1$, $Lit^2$, $Lit^3$, $Lit^4$, $Lit^5$, $Pop^1$, $Pop^2$, $Lit^2Pop^1$, and $Lit^3Pop^1$. These exponent combinations were selected based on examples in the literature and then explored iteratively, stopping at combinations with decreased skill compared to lower order combinations. For each country and exponent combination, the median and the spread of three skill metrics are compared: $\rho$, $\beta$, and RMSF (Fig. 3 and Tables A2 and A3)."*

2.10. Based on the redundancy of either beta or rho (stated above) and your diverging findings for different skill scores within your validation, I find the final decision to use the downscaling with m=n=1 (line 240) very ill-founded. At this stage I would expect a more thorough and stringent assessment of the different exponents and functionalities(see comment above). Otherwise, the full validation exercise seems redundant.

Response: This point is well taken. The comment connects different comments by both reviewers related to the Sections on validation/evaluation. Indeed, we are not validating but comparing different exponents in the evaluation exercise. Please refer to our response to comment 1.0 by the first reviewer for a broader discussion of the limitations of the evaluation and on the resulting changes regarding the reframing of the comparison as evaluation instead of validation.

Regarding the specific concerns raised by this comment: The aim of the manuscript is to present a global asset exposure data set, document the underlying data and methodology (the so called LitPop methodology), and evaluate the methodology by comparing different variations of the downscaling function and selecting the most appropriate combination of exponents for the production of the globally consistent version of the asset exposure data. We understand that this can seem redundant with regards to the structure of the manuscript. In our actual work flow however, the evaluation was undertaken before the global data set was produced to be able to decide on the actual combination of m and n to use. We understand the concerns related to the evaluation that arise from the finding that **(1)** no other functionalities besides $Lit^mPop^n$ were assessed, **(2)** only a limited set of possible values for the exponent combinations for m and n were compared, and **(3)** m=n=1 does only perform best for the skill metrics $\rho$ and $\beta$, but not for RMSF. We also agree that based on the quantitative evaluation, we could have also argued for another choice. We argue that given the global availability of both nightlight intensity and population data and the positive effect of including population data with regards to

saturation and blooming, the chosen functional form and exponents are an appropriate choice with regards to the purpose of the data set and the methodology.

For this we argue by responding to each of the three points of criticism formulated in the reviewer's comment and a summarizing statement:

Ad (1): The functional form $Lit^m Pop^n$ was selected based on previous studies disaggregating asset exposure value or GDP proportionally to either gridded population data (here: $Pop^1$), higher exponents of (here: $Lit^m$ with m>1) or on the product of both data types (i.e. $Lit^1 Pop^1$). Please refer to our response to comment 5 by the same reviewer for more details. The aim of the whole exercise presented in our manuscript was to apply these well-tried approaches within a globally consistent methodology and evaluate varied combinations of *Lit* and *Pop* in accordance with previous applications. While the comparison with different functional forms for combining the two input data types would be beneficial, such a meta study would go beyond the scope of this study.

Ad (2): The range of exponents considered for evaluation was determined based on literature and an iterative approach. Please refer to the responses to the question asked in comment 1.10 by Anonymous Reviewer 1: *"Why did authors choose to vary Lit more than Pop?"* for more details.

Ad (3): Regarding the partial redundancy of $\rho$ and $\beta$, please refer to the response to comment 7. As we will state more clearly in the revised manuscript (see changes below). Regarding the deviating results between $\rho$ and $\beta$ on the one hand and RMSF on the other hand: We chose to consider all three skill-metrics because they convey complementary information. $\rho$ and $\beta$ are computed based on covariance and standard deviation which are representing the distribution of the total values of the input variables (here: nGRP). While they are relatively susceptible to outliers (i.e. putting larger weight on larger values), they represent the distribution of total values. RMSF, on the other hand, weights the relative deviation between modelled and reference nGRP equally, independently of its total value. Prioritizing a better distribution of total values over relative performance, we conclude that $Lit^1 Pop^1$ can be considered the most adequate combination of *Lit* and *Pop* for the subnational downscaling of GDP. Still, we decided to include RMSF in the evaluation sections in order to be transparent regarding the fact that there is no combination of exponents available that performs best with regards to all skill metrics.

For a clearer statement of the results of the evaluation, we added the following in Section 3.2. of the revised manuscript:

*"Within the set of combinations exclusively based on Lit (n=0), the skill metrics $\beta$ and RMSF perform best for $Lit^4$ (Fig. 3b,c), with median $\rho$ improving for larger values of m, however changing little from $Lit^4$ to $Lit^5$ (Fig. 3a).*

*Based on the comparison of the disaggregation skill with varying exponent m and n, there are two candidates for the most adequate functionality: $Lit^1 Pop^1$ (best $\rho$ and $\beta$) and $Lit^4$ (best RMSF and best performance for n=0). The skill metrics of linear regression, $\rho$ and $\beta$, give a better representation of the disaggregation skill for the absolute values than RMSF which is based on the relative deviation per data point. Prioritizing a better distribution of total values over relative performance, we conclude that $Lit^1 Pop^1$ can be considered the most adequate combination of Lit and Pop for the subnational downscaling of GDP. For countries with a lack of highly resolved population data, alternative data sets could be produced based on $Lit^4$ alone."*

2.11. Line 187 (and others in the following): the notion of economically strong (or large)and weak regions is not very well defined. The reader can sort of understand what the authors hint at but it remains very unclear. How do they distinguish strong from weak regions? What is the precise criterion? Does this hold nationally or internationally?

Response: We would like to thank the anonymous reviewer for pointing out that ambiguity. With economically strong/weak regions, we referred to the GRP of the region relative to the other regions in the same country. The differentiation between regions thus holds only nationally. This is the relevant scope for evaluation, as the skill metrics are computed separately for each country.

For clarification, we changed the wording in Section 2.6 (formerly Section 2.7) to:

*"regions with relatively large/low GRP"*

Particular change (new in *blue*):

*"$\beta$ larger (lower) than 1 implies an overestimation (underestimation) of the GRP of  regions with relatively large GRP and an underestimation (overestimation) of  regions with relatively low GRP by  the downscaling within one country."*

In Section 3.4 (formerly 3.3) regarding the example in Mexico, we changed the wording accordingly, now referring to *"districts with relatively low GRP"* instead of *"smaller districts"*.

2.12. The sentence 'There is probably a lot of housing and infrastructure in suburban México that is used by a population that works in the city and thus contributes to the GRP of Mexico City' (strange comparison of stocks and flows) and the following discussion is very difficult to digest for the non-expert reader. I find this discussion very relevant and think it should be extended here or at some other point in the manuscript as it directly links to many relevant issues: a) What does nightlight intensity actually capture? Assets or GDP? b) What is the highest downscaling resolution one should aim at when population is most likely a better proxy of the location of assets but night-light also captures economic activity (e.g. driving cars)? Also in the light of above sentence when GDP and assets seem to be separated by municipal boundaries. c) how can the interaction of both data sources most efficiently be combined? How does the present methodology add to this discussion? How can the different exponents be interpreted in this respect?

The questions raised by the reviewer are indeed very valid and worth to research in greater detail. We agree that the discussion we have offered in the manuscript so far is a stub and would require further explorations. However, a detailed analysis of the case study would be beyond the scope of the paper. The aim of the paper is not to research into questions like these on that level. As a consequence, we changed the structure of the whole Results section (putting more focus on the main results, i.e. the data set) and propose to remove the rudimentary interpretation in the Results section on Mexico (formerly Section 3.3), including the sentence quoted by the reviewer. Still, we keep a revised version of the Section at the end of the revised Results section. The reason is that the purpose of showing the example of Mexico is to illustrates the limitations of the disaggregation approach and gives some insight into the data behind the evaluation. The revised Section

3.4, as pasted below, presents and discusses the nGRP data closer to what is in the data and refrains from unsupported interpretations:

*"The skill metrics for the subnational disaggregation of GDP in the country Mexico shows low value of $\rho$ compared to most other countries for all tested values of m and n ($\rho$=0.76 for $Lit^1Pop^1$, c.f. Table A2a). The example of Mexico is presented here to illustrate limitations and uncertainties of the disaggregation approach. Figure 5 shows the data behind the evaluation for Mexico, i.e. modelled and reference nGRP for all 32 districts of Mexico. The corresponding plot data can be found in Table S2 as supplementary material. While the LitPop methodology performs well for most of the districts with relatively low GRP, it fails to reproduce reference nGRP for the main (capital) metropolitan region consisting of the districts México and Mexico City (Distrito Federal).*

*The two districts with the largest GRP of the highly centralized country are Distrito Federal (Mexico City district) with a reference nGRP of 17.4% and México district (8.7%), surrounding the Distrito Federal. Asset exposure maps of the metropolitan region are shown in Figure A1 in the Appendix. The disaggregation of GDP underestimates nGRP for Mexico City district while overestimating the value for México for all evaluated combinations of m and n (nGRP for $Lit^1Pop^1$, $Lit^3$, and $Pop^1$ are shown in Figure 5). The overestimation of México district's nGRP indicates that the district has an over-proportional nightlight intensity and population count compared to a relatively low reference nGRP. Both districts combined sum up to modelled nGRP values of 11.2 to 17,6% for $Lit^m$, 20.8% for $Pop^1$, and 26.5% for $Lit^1Pop^1$ (Table S2), the latter agreeing well to a combined reference nGRP of 26.1%."*

Additionally, we are taking these results up in the Discussion (Section 4), by adding the following paragraph to the revised manuscript:

*"The example of Mexico (Section 3.4) illustrates the limitations of the LitPop methodology when it comes to the disaggregation of GDP within a metropolitan area: While the disaggregation of GDP proportional to $Lit^1Pop^1$ nicely reproduces the summed nGRP of the metropolitan area, methodology fails to reproduce the distribution of nGRP between the two districts making up the metropolitan area."*

2.13. The discussion in line 284-292 is very vague as it is very hard to judge for the reader when to apply the authors' recommendation: high-resolution vs. coarsely resolved?, use a higher exponent of nightlights instead...instead to what? Why use exponent n=3 when this was never a potentially recommended value in the validation before? The discussion on auxiliary data should be placed somewhere else.

Response: In the lines the reviewer is referring to, we recommended an exponent for Lit >= 3 for countries with coarsely resolved population data available (as the level of detail in the GPW population data varies between countries). Calling the exponent n instead of m (for $Lit^m$) was a typographical error and we would like apologize for that mistake that might have added to the confusion. The recommendation is based on the evaluation result that $Lit^3$, $Lit^4$ and $Lit^5$ show larger disaggregation skill than $Lit^1$. This is now better documented in the Results Section of the revised paper (c.f. changes to the text in the Results as applied in response to the same reviewer's comment 10). Based on the results and the reviewer's comment on the vagueness of the recommendation, we have revised the lines in the Discussion with a focus on precision, please find the revisions below:

Old: *"For countries without a high-resolution distribution of population in the gridded dataset, an exposure map based on $Lit^m Pop$ ( is equivalent to one based on $Lit^m$ alone. For more locally refined risk assessments, in countries with coarsely resolved population information, we advise to use a higher exponent of nightlights instead, i.e. $n \geq 3$."*

Revised: *"For countries without a high detail level in the population data available, asset exposure based on $Lit^m Pop^n$ is more or less equivalent to one based on $Lit^m$ alone. For regional application in these countries, evaluation results suggest that disaggregation proportional to $Lit^4$ could distribute asset values best in the absence of detailed population data."*

Ad *"The discussion on auxiliary data should be placed somewhere else.":* we moved the discussion on auxiliary data into the paragraph on "scalability and flexibility". It is also reformulated and expanded, as can be seen in our response to comment 2.14 below.

2.14. It is very unfortunate that the validation was (or could) only be conducted for 14 countries and no low-income country. The subsequent application of this method to all countries globally has to be treated with caution. In the present manuscript I am missing a detailed discussion of the reliability of the dataset for specific regions and/or income groups and a discussion of potential workarounds. What is the result of the authors' validation in terms of income groups? Is there any information (e.g. trends with income) that could be valuable for low income countries not treated here? What about very small countries, islands, etc? How could other data sources (e.g. household survey data from the Worldbank) be used to improve the data? What has been conducted with this respect in the literature so far (c.f. following paper and the references cited there: Gunasekera, R., et al. (2015). "Developing an adaptive global exposure model to support the generation of country disaster risk profiles." Earth-Science Reviews 150:594-608.)?

Response: The limitation of the evaluation to 14 countries from the income groups 2-4 was brought up by both reviewers. While these limitations were already communicated in the original manuscript, we agree that the communication and discussion should be expended and cautioning remarks added. Please also refer to our response and revisions in reply to the first reviewers comments 1.0 and 1.15.

As this comment is very dense and raises various relevant issues, some beyond the scope of the present paper, indeed. Please find our direct response to the specific points raised below:

Ad *"The subsequent application of this method to all countries globally has to be treated with caution. In the present manuscript I am missing a detailed discussion of the reliability of the dataset for specific regions and/or income groups and a discussion of potential workarounds.":*

We have revised the second paragraph in the Discussions in order to expand the discussion of the limited scope of the evaluation and its consequences, including possible workarounds (changes in *blue*):

*"Based on globally available input data, the LitPop methodology  can be applied across countries from different continents and income groups without any customization. While the presented data set is not complete, it provides data for 224*

*countries contributing 99.9% of global GDP. Therefore, LitPop-based asset exposure data can be used as a basis for globally comparable economic risk assessments. However, the evaluation of the disaggregation skill of the approach presented here is limited to 14 OECD countries. It should be noted that due to lack of data we were not able to evaluate the method's performance for low income countries (World Bank income group 1). Therefore, the application of the asset exposure data for local assessments in countries within low income groups should be treated with caution. Another caveat to global consistency is the fact that the quality and resolution of the underlying population dataset varies between countries, as discussed in greater detail in the next paragraph. As a consequence of these limitations, asset exposure data should be validated against local data before application for local risk assessments, especially in low income countries."*

Ad *"What is the result of the authors' validation in terms of income groups? Is there any information (e.g. trends with income) that could be valuable for low income countries not treated here? What about very small countries, islands, etc?":*

While this would definitely be a valuable contribution for future studies conducting asset value downscaling, a more detailed analysis of a potential income-group dependency of downscaling functionalities would be beyond the scope of this publication. A comparison of skill metrics between the 14 countries did not show a clear picture in term of differences between income groups.

We thus added the following as an outlook to the Conclusion section (additions in *blue*):

*"Future research and development could focus on the integration of higher resolved population data and other ancillary data sources as they become available globally, and validation of the disaggregated asset exposure values against empirical data. Validation against subnational asset value and empirical asset stock inventories yields the potential to evaluate and further improve the accuracy of asset exposure downscaling, both for global and regional applications. Regional validation could inform the choice of the most appropriate downscaling functionality for different income groups and world regions."*

Ad *"How could other data sources (e.g. household survey data from the Worldbank) be used to improve the data? What has been conducted with this respect in the literature so far (c.f. following paper and the references cited there: Gunasekera, R., et al. (2015). "Developing an adaptive global exposure model to support the generation of country disaster risk profiles." Earth-Science Reviews 150:594-608.)?":*

We would like to thank the reviewer for bringing the key publication by Gunasekera, R., et al. (2015) to our attention.

The exposure data model presented by Gunasekera et al. (2015) overlays different methodologies to represent a gridded asset exposure inventory of a country as comprehensively as possible. Their methodologies include exposure disaggregation, building typology and vulnerability distribution, and asset value determination. Among other data sources, their model includes the disaggregation of GDP proportional to gridded population data as well as national asset value estimates that are downscaled. Unfortunately, the resulting dataset is – in contrast to our method and all results – not available online. With the LitPop methodology, we provide a simplified approach for exposure disaggregation only, excluding building typology and vulnerability distribution. We find that the approaches used by Gunasekera et al. (2015) and also other studies

referenced in our manuscript (e.g. De Bono and Mora, 2014, and Murakami and Yamagata, 2019) support the basic approach of the LitPop methodology while showing how the approach could be refined by combination with other data sources. The LitPop methodology and the provided open-source python module have a lower threshold for reproduction and updates when updated nightlight or population data becomes available. However, the LitPop methodology provides less precision when it comes to the assets of specific sectors and building typology.

In summary, we happily include reference to Gunasekera, R., et al. (2015) in the revised manuscript and briefly discuss the LitPop methodology in with respect to their approach.

Revised first references in the Introduction:

*"Due to the lack of comprehensive asset stock inventories, large scale asset exposure maps are often estimated top-down, using downscaling techniques (De Bono and Mora, 2014; Gunasekera et al., 2015; Murakami and Yamagata, 2019)*

*[…]*

*An alternative methodology to model global asset exposure based on the combination of diverse data sets was presented by Gunasekera et al. (2015). The authors combined data on built-up area, building typologies, and construction cost with sector specific asset data and GDP disaggregated proportional to population density. Unfortunately, the source code and resulting exposure data have not been made publicly available. Reproducing these previously mentioned exposure modelling efforts is beyond the scope of most economic disaster risk assessments and climate change adaptation studies."*

Amendments in the Discussion:

*"Additionally, the asset exposure data could be further refined by including auxiliary data, such as road networks and land cover (Geiger et al., 2017; Murakami and Yamagata, 2019), or mobile phone cell antenna density (Brönnimann and Wintzer, 2018). In order to include sector specific assets not represented by the LitPop methodology, i.e. power plants or mines in unpopulated areas, additional sector specific asset inventories should be included (Gunasekera et al., 2015). For a globally consistent approach, sectoral data should however be included with caution, as such datasets are prone to regional or national biases."*

2.15. The concept of intermediate downscaling appears in line 257 very ad-hoc and is used thereafter without further explanation.

Response: By "intermediate downscaling" we refer to the idea of disaggregating total asset value to subnational administrative units proportional to their GRP before disaggregation to grid level. This approach could potentially mitigate geographical biases within countries with large internal structural differences. We initially mentioned this approach here because the functionality is implemented in the LitPop-module of the CLIMADA repository (https://github.com/CLIMADA-project/climada_python/blob/master/climada/entity/exposures/litpop.py). However, we agree with the reviewer that the mentioning in the manuscript is confusing. Since intermediate downscaling is not used for the data set presented, the mentioning is not required.

Changes: We therefore removed both appearances/discussions of the term in the revised manuscript.

2.16. LitPop as a top-down approach is first introduced in line 302. It would make much more sense to make this statement much earlier otherwise one should avoid this notion in general.

Response: We initially introduced the term "top-down" in the discussion to mark the difference to more local (bottom-up) or sector specific approaches to create asset exposure data, i.e. with mapping on the ground or the integration of industry data bases. We agree with the reviewer that this late introduction of the term leads to more confusion than clarification and is not needed to communicate the intended differentiation. Thus, we replaced the term "top-down approach" in the revisions, mainly by the better introduced concept of "disaggregation". We aligned the terminology throughout the manuscript to improve readability, using the term disaggregation more consistently.

In the particular paragraph in the Discussion (Section 4) the reviewer is referring to, we revised the text accordingly and remove the unnecessary remark on the "top-down" nature of the approach (changes in *blue*):

*"Since the CLIMADA repository is open-source, the LitPop methodology can  be amended to include alternative data sources and versions of both gridded nightlight, population,  and total asset values or other socioeconomic indicators to expand and update the  asset exposure data. The LitPop methodology was developed to provide  globally consistent asset exposure  data for global-scale  physical risk modelling. While it could be used for other applications as well, the limitations of its scope should be noted:  The LitPop methodology does not account for differences in infrastructure types and vulnerability. In addition, gridded data may cause poor scoping of areas most vulnerable , or those with more exposed population. [...]  The use for local  or sector specific applications  is limited  without the addition of sector specific data sets."*

2.17. The term 'exposure' is used differently throughout the manuscript. It seems that he authors use it for 'asset exposure' but this is not fully clear. Exposure is very general and could be understood as population or GDP exposure as well. Therefore, I encourage the authors to be more precise and use the expression 'asset exposure' every time they mean it.

Response: We have amended the text as proposed by the reviewer, adding the term *"asset"* for clarification in front of many different appearances of *"exposure"* (most prominently in the title of the manuscript: *"Asset exposure data for global physical risk assessment"*)

The single changes in the text are not listed here, as they are spread through the whole manuscript.

2.18. All abbreviations (e.g. GDP, GRP), all variables, and all subscripts (e.g. pix) need to be explained at first use, even if the authors think that they are self-explanatory. Thereafter another redefinition should be avoided and the authors should stick to their abbreviations.

Response: We have revised the first appearances of following abbreviations, variables, and subscripts based on the reviewer's comment: *GDP, GRP, IQR, $\rho$, $\beta$, RMSF, pix*.

The single changes in the text are not listed here, as they are spread through the whole manuscript.

2.19. Figure 4: The usage of Mexico (country) and México (region) is very confusing fort he reader. Clearly state this difference and maybe use 'México region' to underline the difference.

Response: We would like to thank the reviewer for pointing out the confusion caused by the district names in Mexico. We would suggest to adapt the reviewer's suggestions to use the term "México district" instead of "México region" or just "México". All mentions of México (also in Figure 4) were adapted accordingly in the revised manuscript.

Revisions in Section 3.4 of the revised manuscript (changes marked in *blue*):

" The skill metrics for the subnational disaggregation of GDP in the country Mexico shows low  values of $\rho$ compared to most other countries for all tested values of m and n ($\rho$=0.76 for Lit[1]Pop[1], c.f. Table A2a). […] Figure  4 shows the data behind the evaluation for Mexico, i.e. modelled and reference nGRP for all 32 districts of Mexico. The corresponding plot data can be found in Table  S2 as supplementary material. While the LitPop methodology works well for most of the  districts with relatively low GRP, it fails to reproduce the nGRP for the main (capital) metropolitan region consisting of the districts México and Mexico City (Distrito Federal)."

Further down in the text we consistently replace ”México” with ”México district”.

*Minor points:*

2.20. The discussion in line 31 should include another freely available gridded GDP dataset: Kummu, M., et al. (2018). "Gridded global datasets for Gross Domestic Product and Human Development Index over 1990-2015." Sci Data 5: 180004.

Response: The reference was added as suggested by the reviewer. The cumulated changes to the paragraph are shown below in the response to comment 2.22.

2.21. The reference Murakami et al is outdated. Please update to: Murakami, D. and Y. Yamagata (2019). "Estimation of Gridded Population and GDP Scenarios with Spatially Explicit Statistical Downscaling." Sustainability 11(7).

Response: The reference was updated accordingly.

2.22. Line 34: The statement on high-resolution GDP data availability for academic purposes only is not true. Upon checking the reference I found that the data is freely available. The corresponding reference should be included in the manuscript: Geiger, Tobias; Daisuke, Murakami; Frieler, Katja; Yamagata, Yoshiki (2017): Spatially-explicit Gross Cell Product (GCP) time series: past observations (1850-2000) harmonized with

future projections according to the Shared Socioeconomic Pathways (2010-2100).GFZ Data Services. http://doi.org/10.5880/pik.2017.007

Response: Thank you for pointing out that the data has been made publicly available. We were not aware of the publication of the GCP dataset under a creative commons (CC) 4.0 license when we first submitted the manuscript. Based on the reviewer's comments 2.20 to 2.22, we have updated the reference and the statement on availability as follows (revised version of manuscript text shown in *blue*):

Old (Lines 34-37): *"Assuming that human presence and activity are proxies of economic output, downscaling of gross domestic product (GDP) has been based on population combined with land-use, road networks, and locations of airports (Murakami and Yamagata, 2016). While high resolution yearly GDP maps based on this approach were created for academic use (Frieler et al., 2017), there is no recent global high-resolution exposure dataset available for unrestricted use known to us.*

Revised: *"Assuming that human presence and activity are proxies of economic output, downscaling of GDP has been based on geographical population data (Kummu et al., 2018) and on population combined with land-use, road networks, and locations of airports (Murakami and Yamagata, 2019). High resolution yearly GDP maps based on these approaches are publicly available (Geiger et al., 2017; Kummu et al., 2018)."*

2.23. Line 55 (and others): the reference to Zhao et al. cannot be found in the list of references.

Response: We would like to thank the referee for pointing out the error in this crucial reference. The reference was mistakenly listed under "Naizhuo Zhao" instead of "Zhao, N.". We have corrected this mistake in the revised manuscript.

2.24. Line 177: What does the exponent '5' stand for in nGRP_i? Looks like a footnote which I am unable to locate. Same issue in line 189 and 228.

Response: These superscript numbers are indeed remnants of footnotes that existed in an earlier draft. We would like to apologize for the confusion. The superscript numbers are removed in the revised manuscript.

2.25. Line 182: Seems like the separated equation for rho got lost and appears inline now. The enumeration eq. 4 is also missing.

Response: Equation numbers and references were revised according to the reviewer's observation. The equation for rho is eq. 4.

2.26. Figure 2: I do not understand what do you mean by log-normal colorbar? I would appreciate the colorbar to have a label. What kind of USD do you use here? PPP-adjusted, current or real? This applies similarly for Fig A1.

Response: The description of the colorbar as "log-normal" was indeed incorrect. The colorbar shows disaggregated asset values in current USD of 2014 on a logarithmic scale. This information was added to the caption in the revised manuscript. We have added a label to the colorbar of the figures, as suggested by the reviewer. Additionally, we have

added the information that it is current USD to the captions of Figures 2 and A1. Revised caption:

*"Figure 4: Maps of disaggregated asset exposure value. Values are spatially distributed proportional to nightlight intensity of 2016 (Lit[1], a), population count as of 2015 (Pop[1], b), and the product of both (Lit[1]Pop[1], c) for metropolitan areas in the United Kingdom (GBR) and India (IND). The maps are restricted to the wider metropolitan areas of London (0.6°W-0.4°E; 51-52°N) and Mumbai (72-73.35°E; 18.8-19.4°N) respectively. The colorbar shows asset exposure values in current USD of 2014 per pixel of approximately 1 km$^2$."*

2.27. Line 219: replace top -> bottom

Response: The word was replaced as suggested.

2.28. Line 326-328: The information on RMSF is repeating what the authors mentioned earlier around line 190.

Response: This comment is most likely referring to Line 236-238 (not 326-328). To remove the redundancy, the information on RMSF were removed in the Results section. The following sentence was moved to the Methods Section where RMSF is first introduced, since it is not redundant with the information provided already:

*"A RMSF-value of 2 means that on average, the modelled GRP deviates by a multiplicative factor of 2 from the reference value."*

2.29. Line 240: remove 'an'

Response: The 'an' was removed.

2.30. Line 243: A reference to the data in the appendix would be very helpful here as the reader is unable to extract the information for Mexico from section 3.2.

Response: The paragraph was rewritten for more clarity. As part of this, we added a reference to Table A2a in the Appendix. Lines 242-247 were replaced by the following, as shown in response to comments 2.12. and 2.19 in greater detail:

*"The skill metrics for the subnational disaggregation of GDP in the country Mexico shows low values of $\rho$ compared to most other countries for all tested values of m and n ($\rho$=0.76 for Lit[1]Pop[1], c.f. Table A2a). [...] Figure 5 shows the data behind the evaluation for Mexico, i.e. modelled and reference nGRP for all 32 districts of Mexico. The corresponding plot data can be found in Table S2 as supplementary material. [...] Asset exposure maps of the metropolitan region are shown in Figure A1 in the Appendix."*

2.31. Line 264: the reference for Pittore et al cannot be found in the list of references.

Response: Again, we would like to thank the referee for pointing out the error in this crucial reference. The reference was mistakenly listed under "Massimiliano Pittore" instead of "Pittore, M.". We have corrected this mistake in the citation catalogue for the revised manuscript.

2.32. Line 334: replace get > become

Response: The word was replaced as suggested.

2.33. Caption figure A1: replace 'the Mexico and USA' > 'Mexico and the USA'

Response: The position of the article "the" was changed as suggested.

**References:**

Aznar-Siguan, G. and Bresch, D. N.: CLIMADA v1: a global weather and climate risk assessment platform, Geoscientific Model Development, 12(7), 3085–3097, doi:https://doi.org/10.5194/gmd-12-3085-2019, 2019.

Gettelman, A., Bresch, D. N., Chen, C. C., Truesdale, J. E. and Bacmeister, J. T.: Projections of future tropical cyclone damage with a high-resolution global climate model, Climatic Change, 146(3–4), 575–585, doi:10.1007/s10584-017-1902-7, 2017.

Gunasekera, R., Ishizawa, O., Aubrecht, C., Blankespoor, B., Murray, S., Pomonis, A. and Daniell, J.: Developing an adaptive global exposure model to support the generation of country disaster risk profiles, Earth-Science Reviews, 150, 594–608, doi:10.1016/j.earscirev.2015.08.012, 2015.

Sutton, P., Elvidge, C. and Ghosh, T.: Estimation of gross domestic product at sub-national scales using nighttime satellite imagery, International Journal of Ecological Economics & Statistics, 8(S07), 5–21, 2007.

Sutton, P. C. and Costanza, R.: Global estimates of market and non-market values derived from nighttime satellite imagery, land cover, and ecosystem service valuation, Ecological Economics, 41(3), 509–527, doi:10.1016/S0921-8009(02)00097-6, 2002.

Zhao, N., Samson, E. L. and Currit, N. A.: Nighttime-Lights-Derived Fossil Fuel Carbon Dioxide Emission Maps and Their Limitations, Photogram Engng Rem Sens, 81(12), 935–943, doi:10.14358/PERS.81.12.935, 2015.

Zhao, N., Liu, Y., Cao, G., Samson, E. L. and Zhang, J.: Forecasting China's GDP at the pixel level using nighttime lights time series and population images, GIScience & Remote Sensing, 54(3), 407–425, doi:10.1080/15481603.2016.1276705, 2017.